# MYT1L deficiency impairs excitatory neuron trajectory during cortical development

Allen Yen [1,2], Simona Sarafinovska[1,2], Xuhua Chen[1,3], Dominic D. Skinner [4], Fatjon Leti[4], MariaLynn Crosby[3,5], Jessica Hoisington-Lopez[3,5], Yizhe Wu [1,2], Jiayang Chen [1,2], Zipeng A. Li [1,2], Kevin K. Noguchi[2], Robi D. Mitra[1,3,7] & Joseph D. Dougherty [1,2,6,7] ✉

Mutations reducing the function of MYT1L, a neuron-specific transcription factor, are associated with a syndromic neurodevelopmental disorder. MYT1L is used as a pro-neural factor in fibroblast-to-neuron transdifferentiation and is hypothesized to influence neuronal specification and maturation, but it is not clear which neuron types are most impacted by MYT1L loss. In this study, we profile 412,132 nuclei from the forebrains of wild-type and MYT1L-deficient mice at three developmental stages: E14 at the peak of neurogenesis, P1 when cortical neurons have been born, and P21 when neurons are maturing, to examine the role of MYT1L levels on neuronal development. MYT1L deficiency disrupts cortical neuron proportions and gene expression, primarily affecting neuronal maturation programs. Effects are mostly cell autonomous and persistent through development. While MYT1L can both activate and repress gene expression, the repressive effects are most sensitive to haploinsufficiency, likely mediating MYT1L syndrome. These findings illuminate MYT1L's role in orchestrating gene expression during neuronal development, providing insights into the molecular underpinnings of MYT1L syndrome.

Every brain cell shares the same genetic code, yet they exhibit a wide range of functions. This diversity arises because different cell lineages enact different gene expression programs that direct each cell in the embryonic brain to develop in a highly orchestrated manner. Disruption of these processes can lead to abnormal neurodevelopment and result in impaired cognition, communication, and adaptive behavior, as seen in profound autism and intellectual disability (ID)[1,2]. Notably, many genes associated with such neurodevelopmental disorders (NDDs) are expressed early during brain development and are involved in gene regulation and synaptic function[3,4]. Studies using post-mortem human brain tissue provide evidence that cortical excitatory neurons are commonly dysregulated in autism[5,6]. However, since these are end of life studies, whether this is a cause or consequence of autism is unclear.

One NDD associated gene is Myelin Transcription Factor 1 Like (MYT1L), which is highly expressed exclusively in postmitotic neurons in the embryonic brain and sustained at lower levels throughout life[7,8]. Early fibroblast-to-neuron transdifferentiation studies demonstrate that MYT1L promotes neuronal cell fate by repressing non-neuronal lineage programs[9,10]. Similarly, in vivo, epigenetic studies of normal development show that MYT1L promotes neuronal differentiation by recruiting the SIN3B repressive complex to promoters and enhancers of postmitotic neurons to suppress early developmental programs[11]. Indeed, loss of MYT1L in multiple mouse models resulted in upregulation of a fetal gene expression signature[12–14]. To date, three pivotal studies have investigated the in vivo functions of MYT1L by creating transgenic mouse models. Each study disrupted a different exon of

[1]Department of Genetics, Washington University School of Medicine, Saint Louis, MO, USA. [2]Department of Psychiatry, Washington University School of Medicine, Saint Louis, MO, USA. [3]Edison Family Center for Genome Sciences and Systems Biology, Washington University School of Medicine, Saint Louis, MO, USA. [4]Scale Biosciences, San Diego, CA, USA. [5]DNA Sequencing and Innovation Lab, Washington University School of Medicine, Saint Louis, MO, USA. [6]Intellectual and Developmental Disabilities Research Center, Washington University School of Medicine, Saint Louis, MO, USA. [7]These authors jointly supervised this work: Robi D. Mitra, Joseph D. Dougherty. ✉e-mail: jdougherty@wustl.edu

MYT1L (6 in Wohr et al.[15], 9 in Kim et al.[13], and 11 in Chen et al.[12]). The animal models are valuable tools to study the molecular and cellular consequences of MYT1L haploinsufficiency and the mice recapitulate many of the clinical presentations such as hyperactivity, structural malformations, obesity, and behavioral deficits[12–14]. However, it remains largely unknown how MYT1L haploinsufficiency influences the trajectory of neuronal differentiation in vivo, and whether the development of specific neuronal subtypes is particularly susceptible to the loss of MYT1L. Moreover, it is unclear if there is a critical moment in each cell's developmental window during which MYT1L function is indispensable. Understanding this timeline could delineate when the transcriptional dynamics and developmental processes are amenable to interventions.

Detailed atlases mapping the gene expression profiles of thousands of cell types across the entire mouse brain have significantly advanced our understanding of brain organization under typical conditions[16–21]. Building upon this foundational knowledge, we can now explore how genetic perturbations affect neurodevelopment, for example by investigating the impact of disrupting a gene regulatory network through the loss of a single TF on this atlas. Given the widespread expression pattern of MYT1L in neurons, it is unclear if specific neuronal subtypes are more sensitive to MYT1L deficiency. Likewise, previous studies using bulk RNA sequencing have shown that MYT1L deficiency affects genes associated with the cell cycle[11,12,14], differentiation[9,10], and proliferation[22]. However, a limitation of bulk sequencing is that it only provides average gene expression data from a mixed population of cells, making it challenging to discern the precise origin of observed differences. For example, MYT1L haploinsufficiency results in an increased expression of developmental gene expression programs in vitro and in the post-natal brain[10,12,14], but it remains unclear whether the observed differences are due to an increased proportion of immature progenitors or whether post-mitotic neurons are generated in proper numbers, but fail to mature completely and become trapped in an intermediate state. Previous studies have shown that MYT1L can function as both a transcriptional repressor[10,23] and activator[12,24]. This raises an important question: does MYT1L's role as a repressor or activator vary depending on cell type or developmental stage? Additionally, how sensitive are the activated or repressed gene targets to disruption? Although loss of MYT1L leads to precocious differentiation during development[12] and sustained activation of developmental programs in the adult brain[10,11], the implications for neuronal development trajectory and cell-type specific fate specification remain unknown. Utilizing single cell transcriptomics, we can obtain a high-resolution mapping of dynamic developmental processes, potentially elucidating how MYT1L's function differs across various cell types and developmental stages. This approach may also explain the seemingly dual nature of MYT1L as both a repressor and activator and provide insights into how the loss of MYT1L contributes to the observed differential gene expression patterns.

In this study, we profiled a total of 412,132 nuclei to investigate the molecular and cellular consequences of MYT1L haploinsufficiency at the peak of neurogenesis (E14), when neurons in the six cortical layers have been born (P1), and when neurogenesis is complete, and the neurons are maturing (P21). Our findings indicate that MYT1L deficiency primarily impacts excitatory neurons. We further identified that genes regulated by MYT1L, whether activated or repressed, exhibit cell type-specific responses to MYT1L haploinsufficiency. A significant number of dysregulated genes were TFs or epigenetic regulators temporally expressed during specific time windows, highlighting lineage specific gene regulatory networks. In summary, our findings provide insights into how MYT1L haploinsufficiency disrupts embryonic and postnatal neurodevelopment. We have identified key transcriptional networks and defined the vulnerable cell types and developmental stages that potentially contribute to the pathogenesis of MYT1L syndrome.

## Results

### Loss of MYT1L disrupts proportions of excitatory and inhibitory neurons

To characterize the role of MYT1L during peak neurogenesis and to understand the acute consequences of MYT1L haploinsufficiency and loss on cell fate specification and maturation, we applied a massively parallel barcoding approach[25,26] to profile and analyze transcription from 216,830 nuclei from the developing forebrain of embryonic day 14 (E14) MYT1L knockout (KO), heterozygous (Het), and wild type (WT) animals (Fig. 1A, B). We find that cell types are well represented across all genotypes (median genotype LISI score[27] = 2.7) (Fig. 1C–F). We identified 26 clusters representing 7 broad neural cell types, which were further classified into three subtypes of radial glial cells (*Hes1* and *Nestin* positive), 3 subtypes of intermediate progenitor cells (*Neurog2* and *Eomes* positive) fated to be excitatory neurons, 3 subtypes of inhibitory intermediate progenitor cells (*Dlx1* and *Nkx2.1* positive), 8 subtypes of excitatory neurons (*Neurod6* and *Tbr1* positive), 9 subtypes of inhibitory neurons (*Gad1* and *Gad2* positive), Cajal-Retzius cells, oligodendrocyte progenitor cells, and microglia (Fig. 1C–G). We assigned cell cycle scores based on cell cycle phase marker gene expression and confirmed that the progenitors were mostly in G2M or S, while the post-mitotic neurons were in G1/G0 (Fig. 1H) and expressed MYT1L (Fig. 1I). The progenitor cells segregated into two distinct populations, which gave rise to divergent excitatory and inhibitory neuron developmental trajectories. This profile of cellular diversity indicated that we captured a developmental window encompassing differentiation and maturation processes, enabling us to investigate the molecular and cellular consequences of loss of MYT1L in the developing E14 cortex.

Because MYT1L is highly expressed in virtually all neurons during neurogenesis (Fig. 1I), we aimed to assess the short-term consequences of its deficiency on overall cell type proportions. We observed subtle but statistically significant disruptions in the abundance of post-mitotic immature excitatory neurons (Im ExN_3), deep layer excitatory neurons (Im L5-6 ExN_1, Im L5-6 ExN_2, L5-6 ExN_1, L5-6 ExN_2, and Im L6 ExN), immature inhibitory neurons (Im InhN_3), and specific subtypes of inhibitory neurons (somatosensory cortex (SI), Darpp32+ D1-D2, and CEA-BST) (Fig. 1J). The proportions of radial glia (RG) and inhibitory intermediate progenitors (InhIP) were mostly unaffected by the loss of MYT1L. However, Het and KO progenitors from the RG_2 and InhIP_1 clusters showed higher proportions in G0/G1 compared to WT, with a trend towards decreased proportions in the S phase (Fig. 1J). This supports the hypothesis that the loss of MYT1L reduces cell proliferation, disrupting proper cortical development and potentially leading to microcephaly[12]. Non-cycling immature excitatory neurons (Im ExN_3) presumably in the subventricular zone (SVZ) were the most developmentally immature post-mitotic neurons in the excitatory trajectory that showed an increase in abundance in KOs compared to WT. This could be a result of developing neurons prematurely differentiating and making the cell fate decision early at the expense of proliferating progenitors as has been hypothesized previously[12,28].

### Loss of MYT1L disrupts excitatory neuron development

We conducted a differential expression analysis to analyze the molecular signatures of each cell type and determine which subtypes exhibited the most significant transcriptional changes due to MYT1L deficiency. The massively parallel barcoding approach enables multiple biological replicates in a single workflow, helping ensure accurate results and minimize false discoveries. We utilized a pseudobulk analysis, aggregating individual nuclei into cell type groups. This method, recognized for its speed and accuracy compared to specialized single-cell DE methods[29,30], enabled us to perform pairwise analyses between WT and KO genotypes using DESeq2[31]. Our analysis identified 900 unique differentially expressed genes (DEGs; BH adjusted P-value < 0.05; |log2 fold change| > 0.2) between WT and KO, with 600

consistently upregulated and 285 consistently downregulated in KO clusters compared to WT (Fig. 2A, Supplementary Data 1). Notably, deep layer excitatory neurons harbored the majority of DEGs, even after downsampling, indicating their particular sensitivity to MYT1L loss. To ensure that this was not due to differences in sensitivity due to different rates of gene capture, we confirmed the number of DEGs was not correlated with the number of detected genes ($R^2 = 0.07$).

Progenitor cells, which do not express MYT1L, showed no DEGs, suggesting that the effects of MYT1L deficiency are intrinsic to MYT1L-expressing cells.

Given that MYT1L homozygotes do not survive postnatally, and the human disorder is caused by haploinsufficiency, we also analyzed Hets. Pairwise pseudobulk analysis between WT and Het revealed only 5 DEGs across all cell types (Supplementary Data 2). The apparent lack

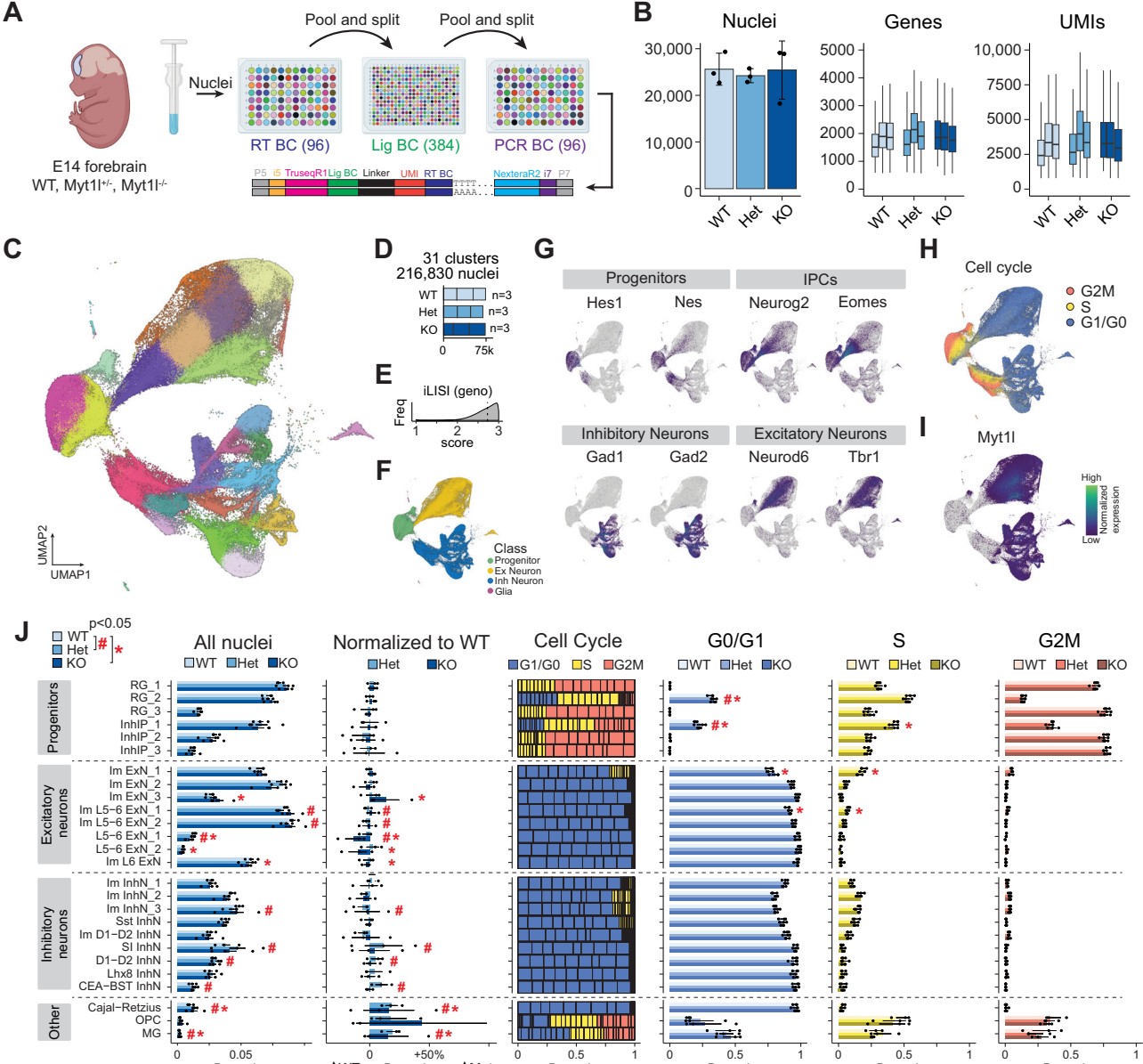

**Fig. 1 | Single nucleus transcriptional profiling of E14 forebrain in MYT1L animals. A** Schematic showing dissection of forebrain tissue, isolation of nuclei, massively parallel barcoding, and generation of snRNAseq libraries. Created in BioRender. Dougherty, J. (2024) BioRender.com/k40k912. **B** General library statistics showing mean ± SD nuclei per genotype (n = 3 biological replicates per genotype: WT, Het, and KO), median ± SD genes per nucleus, and median ± SD UMIs per nucleus. Box plots show median (center line), interquartile range (box), and whiskers extending to 1.5 times the interquartile range. **C** Uniform manifold approximation and projection (UMAP) showing 216,830 nuclei from the forebrain of E14 MYT1L WT (n = 3), Het (n = 3), and KO (n = 3) animals colored by cell type. **D** Bar plot showing the total number of nuclei per genotype across biological replicates. **E** Histogram showing the local inverse Simpson's index (LISI) score with a median of 2.7, indicating that the genotypes are well mixed and integrated. **F** UMAP of all nuclei color-coded by cell class. **G** Top cluster markers for

progenitors, intermediate progenitor cells (IPCs), inhibitory neurons, and excitatory neurons. **H** UMAP of all nuclei color-coded by cell cycle score based on expression of cell cycle genes (G2M phase in orange, S phase in yellow, and G1/G0 in blue). **I** UMAP feature plot showing expression of MYT1L in postmitotic excitatory and inhibitory neurons. **J** From left to right, plots showing: the mean ± SEM relative proportions of nuclei in each annotated cell cluster for MYT1L WT, Het, and KO genotypes (n = 3 biological replicates per genotype); Het and KO mean ± SEM proportions normalized to WT; the proportions of nuclei in the phases of the cell cycle per replicate; mean ± SEM proportions of nuclei that are in the G0/G1 phase; mean ± SEM proportions of nuclei in the S phase; and mean ± SEM proportions of nuclei in the G2M phase. "#" indicates a significant difference between WT and Het, and "*" indicates a significant difference between WT and KO (FDR adjusted p < 0.05, moderated ANOVA).

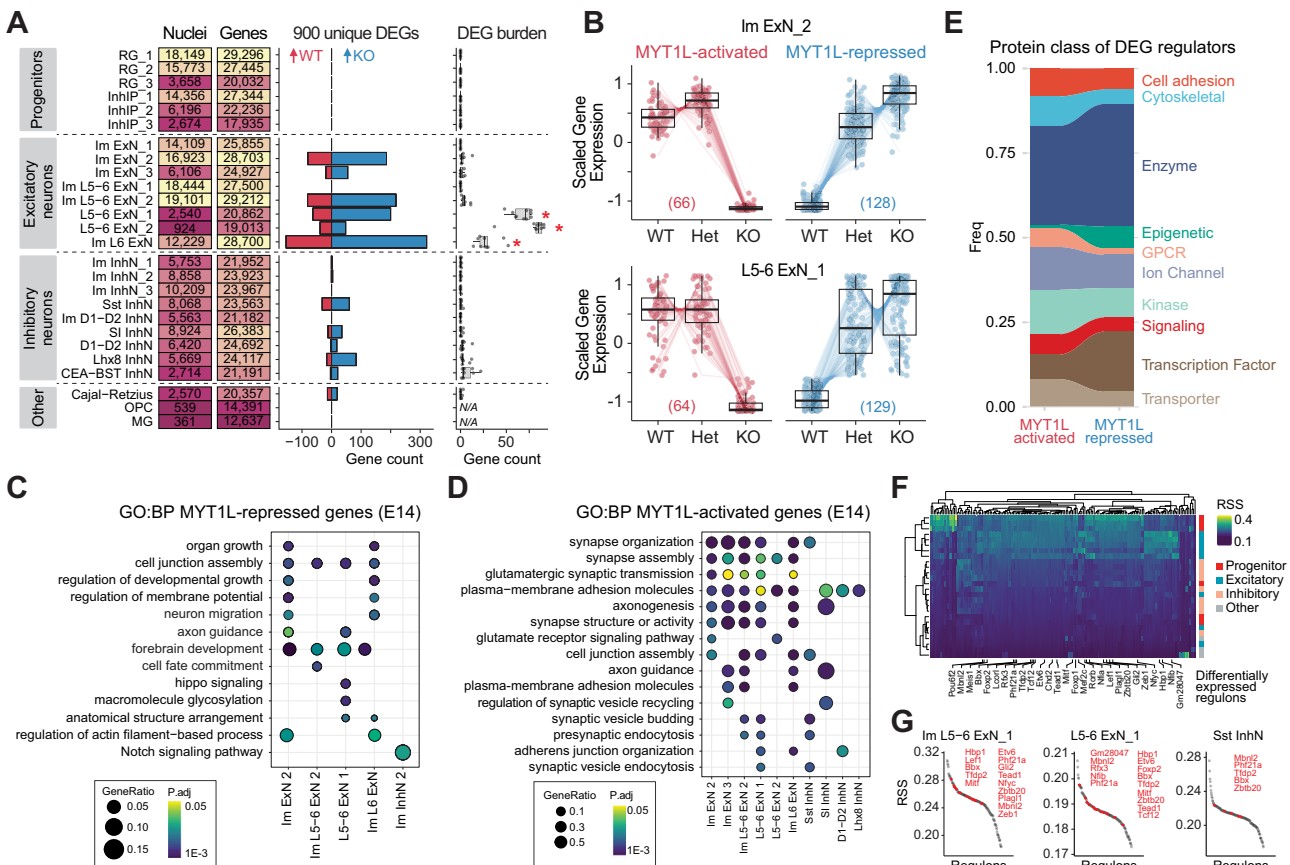

**Fig. 2 | Cell type-specific changes in gene expression at E14. A** Summary plot showing the numbers of nuclei and genes detected in each cluster (left). Bar plot (middle) displays the number of differentially expressed genes (DEGs) upregulated in WT (red) or KO (blue) conditions (n = 3 biological replicates per condition). DEG burden analysis (right) shows the distribution of DEGs per cell type, normalized by the number of nuclei in each cluster. Statistical significance was assessed using a two-sided Mann-Whitney U test with Benjamini-Hochberg correction for multiple corrections (*FDR adjusted p < 0.05). Box plots show median (center line), interquartile range (box), and whiskers extending to 1.5 times the interquartile range. **B** MYT1L gene dose-dependent expression patterns of DEGs in Im ExN_2 and L5-6 ExN_1 clusters, categorized as MYT1L-activated (decreased expression in KO) and MYT1L-repressed (increased expression in KO). Box plots show median, interquartile range, and whiskers extending to 1.5 times the interquartile range. Dot

plots showing enriched GO biological processes terms in MYT1L-repressed (**C**) and MYT1L-activated (**D**) DEGs. Overrepresentation of genes within GO terms was determined using a one-sided hypergeometric test, with p values adjusted for multiple testing using the Benjamini-Hochberg method. **E** Plot showing the frequency distribution of annotated protein classes among the DEGs. **F** Heatmap displaying identified regulons (columns) for each WT cluster (rows), colored by the Regulon Specificity Score (RSS). RSS measures the specificity of each regulon's activity for each cluster. Cluster classes are annotated on the right, and differentially expressed regulons are annotated in each column. **G** Representative plots showing the ranked Regulon Specificity Score plots for Im L5-6 ExN_1, L5-6 ExN_1, and Sst InhN clusters. Differentially expressed regulons are highlighted in red and labeled on the plot.

of DEGs may be due to the subtle magnitude of effect based on previous bulk RNAseq studies[12]. Another possible factor is that E14 is a dynamic differentiation period, where cells are likely in different cell states along the developmental trajectory and not synchronized, resulting in high variability of gene expression within clusters. Thus, rather than a pairwise comparison, we modeled the number of functional alleles as an ordinal factor and applied the Likelihood Ratio Test to evaluate changes as a function of MYT1L gene dose. This can provide insight into whether this pattern of regulation was the same for activated and repressed genes, which may suggest which function is most critical to the disorder. We classified 414 genes most upregulated in KOs as MYT1L-repressed genes, while 232 genes most upregulated in WTs were considered MYT1L-activated genes (Supplementary Fig. 1 and Supplementary Data 3). MYT1L-repressed genes showed greater gene dose sensitivity than MYT1L-activated genes (Fig. 2B), becoming upregulated even with the loss of a single MYT1L allele.

To evaluate if a particular cell type was driving the immature transcriptional signature previously observed in E14 MYT1L Het mouse cortex[12], we performed gene ontology (GO) enrichment analysis for the DEGs in each cluster. MYT1L-repressed genes were enriched in

development, neuron migration, and cell fate commitment pathways, particularly in immature and deep layer excitatory neuronal clusters (Fig. 2C). Conversely, MYT1L-activated genes were involved in synapse organization, axonogenesis, and neurotransmitter secretion and transport (Fig. 2D). Together, this revealed that loss of MYT1L results in an immature developmental transcriptional state, reinforcing that the suppression of developmental genes is critical to ensure proper neuronal maturation.

We next integrated our pseudobulk DEGs with an age and region-matched E14 forebrain MYT1L CUT&RUN dataset that cataloged 560 high-confidence MYT1L binding sites within promoter sequences[11] for 480 genes to identify if the MYT1L-activated and -repressed genes are direct or indirect targets of MYT1L. Of 55 differentially expressed MYT1L targets, 42 were upregulated and 13 were downregulated in KOs compared to WT, reinforcing MYT1L's primary role as a transcriptional repressor. Additionally, differentially expressed MYT1L targets were significantly enriched for transcription factors (TFs) (81/480; $P = 2.2 \times 10^{-16}$, Fisher's exact test) (Fig. 2E), demonstrating that the DEGs were largely driven by indirect effects of MYT1L deficiency.

To gain insight into upstream regulators, we used SCENIC[32] to build a co-expression network and identify regulons, which are modules of putative TF regulators and their inferred target genes. We identified 984 regulons in WT cells, of which 171 regulons were enriched with some cell type-specific activity based on the regulon specificity score (RSS) (Fig. 2F). While MYT1L loss did not significantly disrupt the inferred co-expression network structure, 27 out of 171 (16%) regulons were differentially expressed, primarily in excitatory neurons (Fig. 2G). This provides evidence that MYT1L can be a transcriptional regulator that not only influences its direct targets but also downstream indirect targets within a gene network.

Finally, to examine the convergence of our observed transcriptional disruptions with neurodevelopmental disorder-associated genes, we intersected the excitatory and inhibitory neuron DEGs (Fig. 2A) with 932 high-confidence autism-related genes from the SFARI database with a score of 1 or 2[33–35]. We found a significant overlap of 140 out of 900 DEGs (15.6%) with SFARI genes (P = 2.1 × 10$^{-12}$, chi-square test with Yates' continuity correction), comprising 123 genes from the excitatory neuron clusters and 16 genes from inhibitory neuron clusters. To account for potential neuronal expression bias of autism genes[36], we randomly sampled 900 genes from the top 50% expressed genes in these clusters 1000 times. The median overlap with SFARI genes was 26, compared to our observed 123, indicating a 4.7-fold enrichment for SFARI genes among MTY1L DEGs. Notably, many autism-associated DEGs, such as *Zbtb20* and *Phf21a*, exhibited pronounced dysregulation in deep layer excitatory neurons (Supplementary Fig. 2). This finding suggests that pathways perturbed by MYT1L deficiency share similarities with those disrupted by a subset of autism genes involved in axon guidance, neuronal migration, and chemical synaptic transmission[6]. Overall, these observed transcriptomic changes reveal molecular changes affecting the maturation and function of deep layer excitatory neurons and convergence with key autism-related genes and pathways.

## MYT1L's critical role in neuronal maturation timing

Our differential analysis shows that MYT1L deficiency is associated with an immature transcriptional signature in excitatory neurons. We hypothesize that MYT1L-deficient progenitors undergo premature differentiation, making early neuronal cell fate decisions but subsequently exhibiting delayed or stalled maturation. This results in a disrupted transcriptional maturation signature. We propose a critical moment during differentiation when MYT1L function is essential for guiding the neuronal developmental trajectory. This model reconciles both the precocious differentiation and the immature transcriptional signatures observed in MYT1L-deficient neurons. To test these hypotheses, we assessed maturation trajectory differences between genotypes during the critical developmental window of neurogenesis. Using Monocle3[37], we reconstructed a pseudotemporal trajectory independent of prior cluster definitions (Fig. 3A). This approach models the cell states as a continuum of dynamic changes, enabling quantification of gene expression changes during differentiation. We observed subtle yet widespread disruptions in the distribution of Het and KO nuclei compared to WT across pseudotime states (Fig. 3B–D), suggesting a disrupted transcriptional maturation signature that may be overlooked when examining cell proportions based on cluster markers alone.

To identify drivers of excitatory neuron development, we analyzed TF expression along pseudotime in WT cells, establishing a putative timeline of gene activation and expression from progenitors to differentiated excitatory neurons (Fig. 3E). We then tested whether MYT1L loss causes variations in the timing of TF expression, potentially impacting the developmental trajectory of excitatory neurons. Using the Kullback-Leibler divergence test, we identified 27 TFs with disrupted expression timing as a result of MYT1L deficiency (Fig. 3F, G).

These TFs, generally de-repressed in Hets and KOs, are involved in developmental regulation (*Dlx5*, *Dlx6*, and *Hoxd10*), control of cell cycle progression (*Hbp1*), neurogenesis (*Nhlh2*, *Lmx1a*, and *Insm2*), and epigenetic regulation (*Tet2* and *Prdm*) (Fig. 3G, H). To pinpoint where MYT1L may have the greatest effect, we intersected all the excitatory pseudotemporal TFs with E14 MYT1L CUT&RUN peaks[11]. We found an enrichment of direct MYT1L targets during a transient period shortly after the transition from progenitor to postmitotic neuron, suggesting its important role during this critical moment (Fig. 3F). Additionally, we identified six genes within this pseudotime bin (*Efna4, Ccng2, Nbr1, Frmd4b, Sorsb2*, and *Midn*) as targets of ZBTB12, a molecular gatekeeper known to safeguard the unidirectional transition of progenitors to differentiated states[38]. This analysis provides a pseudotime-resolved sequence of MYT1L target gene expression and identifies a critical developmental window, where alterations in gene expression patterns during this sensitive period may lead to disruptions in neuronal differentiation and maturation.

## MYT1L loss disrupts early postnatal cortical development at P1

To investigate the effects of MYT1L deficiency on neuronal development after all cortical neurons are born, we performed a snRNAseq analysis on the forebrain tissue from P1 MYT1L WT (n = 4) and Het (n = 8) animals. In our MYT1L mouse model, KO animals are not viable postnatally so we are only able to analyze WT and Hets. We analyzed 98,797 nuclei, identifying 36 distinct cell clusters (Fig. 4A). Analysis of cluster proportions revealed significant decreases in deep layer excitatory neurons (L5-6 ExN_2 and L6 ExN), upper layer excitatory neurons (Im L2-4 ExN_2 and L2-4 ExN_2), and increase in Im L2-4 ExN_1 in Hets compared to WT. Interestingly, pseudobulk differential analysis identified 89 unique DEGs (Fig. 4B, Supplementary Fig. 3 and Supplementary Data 4), a marked decrease from the number of E14 DEGs comparing KO to WT, but more than Het to WT comparisons at E14, suggesting an increasing disruption with time. The majority of these DEGs were upregulated in Hets, consistent with MYT1L's primary role as a transcriptional repressor.

To understand the functional implications of these transcriptional changes, we performed GO enrichment analysis on MYT1L-activated and -repressed genes. MYT1L-activated genes, which were downregulated in Het animals, were enriched for processes crucial for neuronal function and maturation including cell-cell adhesion, synaptic transmission, and glutamate receptor signaling (Fig. 4C). This suggests that loss of MYT1L leads to impaired neuronal maturation and synaptic function. Conversely, MYT1L-repressed genes, which were upregulated in Het animals, were associated with developmental processes, including neuron projection guidance (Fig. 4D). This indicates that MYT1L deficiency results in the persistence of developmental programs.

Next, we performed a pseudotime analysis on the continuous excitatory neuron clusters to test for shifts in the distribution of cells per genotype within each pseudotime bin (Fig. 4E). At P1, we detected altered distributions primarily within the upper layer neuron clusters, with Het cells trending towards lower pseudotime values. Similarly to E14, these findings suggest an immature neuronal state compared to WT cells (Fig. 4F). These results further suggest that WT cells mature along the developmental trajectory, while Hets cells lag behind (Fig. 4G), reinforcing the role of MYT1L in proper neuronal development.

Overall, MYT1L likely plays a crucial role in regulating the proportions of maturation of specific neuronal subtypes during early postnatal development, consistent with previous studies[39]. The observed changes in cell type proportions and gene expression patterns suggest that MYT1L deficiency leads to persistent immature transcriptional states and altered neuronal composition, which may contribute to the neurodevelopmental phenotypes associated with MYT1L syndrome.

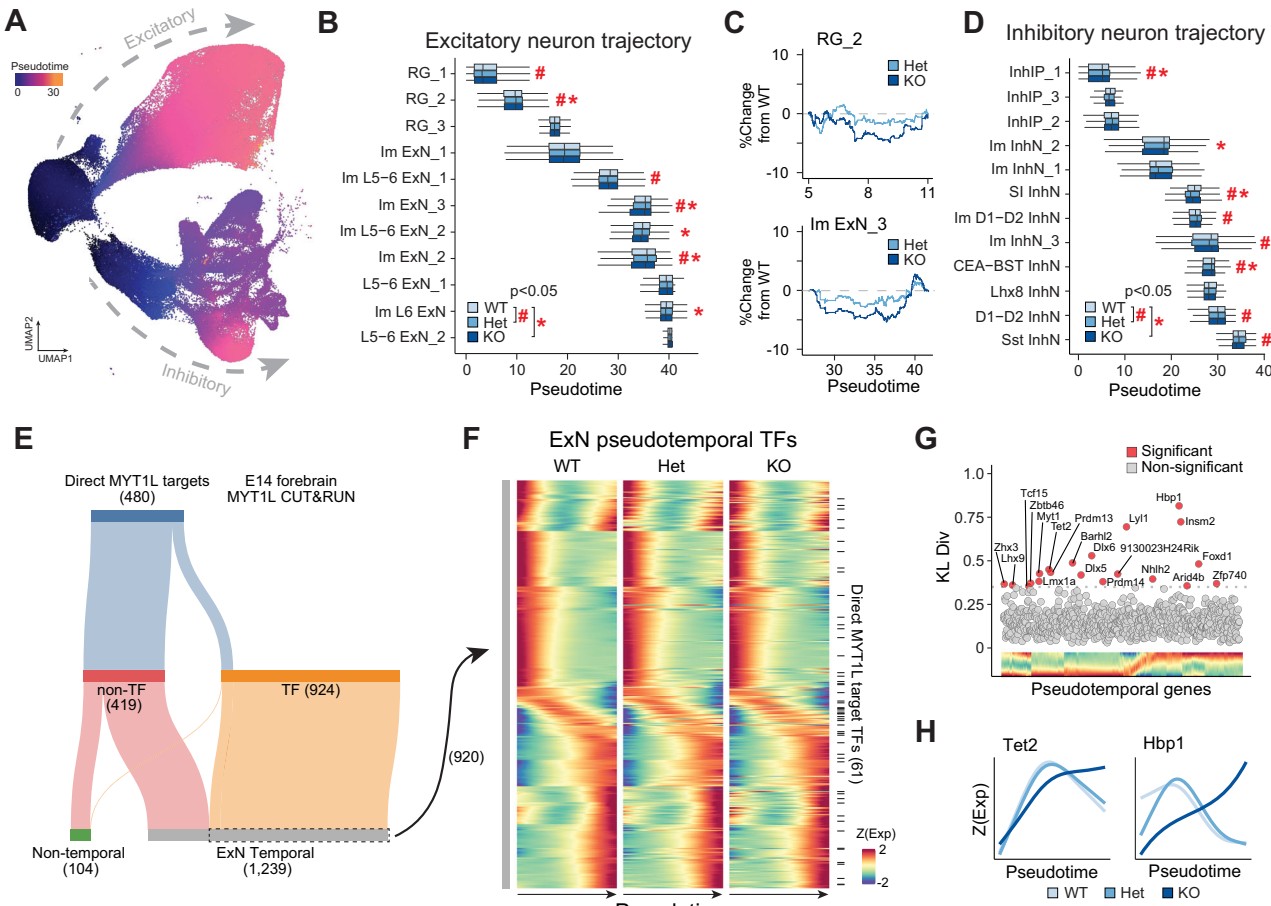

**Fig. 3 | Loss of MYT1L disrupts excitatory neuron maturation at E14. A** UMAP of all E14 nuclei colored by pseudotime. **B** Box plots showing distributions of nuclei from excitatory neurons along pseudotime per genotype (n = 3 biological replicates per genotype: WT, Het, and KO). "#" indicates a significant difference between WT and Het distributions, and "#" indicates a significant difference between WT and KO distributions (FDR adjusted p < 0.05, two-sided Kolmogorov-Smirnov test). Box plots show median (center line), interquartile range (box), and whiskers extend to 1.5 times the interquartile range. **C** Representative plots showing the relative differences in distributions of MYT1L Het and KO nuclei compared to the WT distribution in the RG_2 and Im ExN_3 clusters. **D** Box plots showing the distributions of inhibitory neuron nuclei along pseudotime per genotype (n = 3 biological replicates per genotype: WT, Het, and KO). "#" indicates a significant difference between WT and Het distributions, and "*" indicates a significant difference between WT and KO distributions (FDR adjusted p < 0.05, two-sided Kolmogorov-Smirnov test). Box plots show median, interquartile range, and whiskers extending to 1.5 times the interquartile range. **E** Diagram displaying the breakdown of direct MYT1L targets identified by CUT&RUN that were transcription factors and showed gene expression changes across pseudotime in the excitatory neuron trajectory. **F** Heatmaps showing scaled expression of WT (left), Het (middle), and KO (right) excitatory neuron pseudotemporal genes. Each row represents a gene, with rows sorted according to their expression peak in pseudotime. Black tick marks on the right indicate rows where genes are MYT1L direct targets as determined by CUT&RUN in (**E**). **G** Scatterplot showing the Kullback-Leibler divergence metric used to identify differential pseudotemporal expression profiles in KOs compared to WT. **H** Representative traces of the differential pseudotemporal gene expression profiles for *Tet2* and *Hbp1* across genotypes.

## Sensitivity of excitatory neurons persist throughout neurodevelopment

Finally, to investigate the long-term effects of MYT1L deficiency on both cell proportion and transcriptional changes, we performed snRNAseq analysis of cortical tissue from juvenile WT (n = 6) and MYT1L Het (n = 6) animals at P21, when neurogenesis is largely complete, but critical periods in cortical circuit maturation are ongoing. Analysis of KOs was again not possible due to postnatal lethality[12–14]. We analyzed 96,505 nuclei, identifying 19 types of excitatory neurons spanning cortical layers, 11 subtypes of inhibitory neurons, and 8 non-neuronal types using hierarchical correlation mapping and referencing the taxonomies and subclass annotations from the Allen Brain Cell (ABC) Atlas[19] (Fig. 5A and Supplementary Fig. 4). Analysis of excitatory neuron proportions, revealed significantly fewer upper layer L2/3 IT ENT and L4/5 IT neurons in MYT1L Het cortices compared to WT, with increased numbers of deep layer L6 CT, L6 IT, and L6b CT neurons (Fig. 5B). This is consistent with increased deep layer neuron density in the cortices of P60 Het mice[11]. We detected 412 unique cluster

pseudobulk DEGs, primarily in excitatory neurons (Fig. 5B, Supplementary Fig. 5, and Supplementary Data 5). Consistent with E14 and P1 findings, MYT1L Het neurons showed an increased number of upregulated DEGs, indicating de-repression. We observed a progressive increase in DEGs from E14 (5 DEGs) to P1 (89 DEGs) to P21 (412 DEGs) in Hets compared to WT. L6 neurons were the most affected, with modest effects on L2/3 intrathalamic (IT) and L4/5 IT neurons. GO analysis revealed that WT-upregulated DEGs were associated with axon guidance (*Epha3*, *Epha5*, *Epha6*, *Epha7*, *Slit2*, and *Robo2*), synapse organization (*Cdh6*, *Cdh9*, *Sema3a*, *Cntn5*, and *Lrrc4c*), and neurotransmission (*Gria4*, *Glra2*, *Rim1*, and *Grm3*) (Fig. 5C), while Het-upregulated genes were enriched in nervous system development pathways (*Pak1*, *Numbl*, *Etv5*, *Smurf1*, *Bcl2*, *Nfix*, and *Ptn*) (Fig. 5D). This analysis suggests that MYT1L Het excitatory neurons remain transcriptionally immature compared to WT, demonstrating that neurodevelopmental deficiencies in Hets commence during embryonic development and persist and expand into early postnatal development.

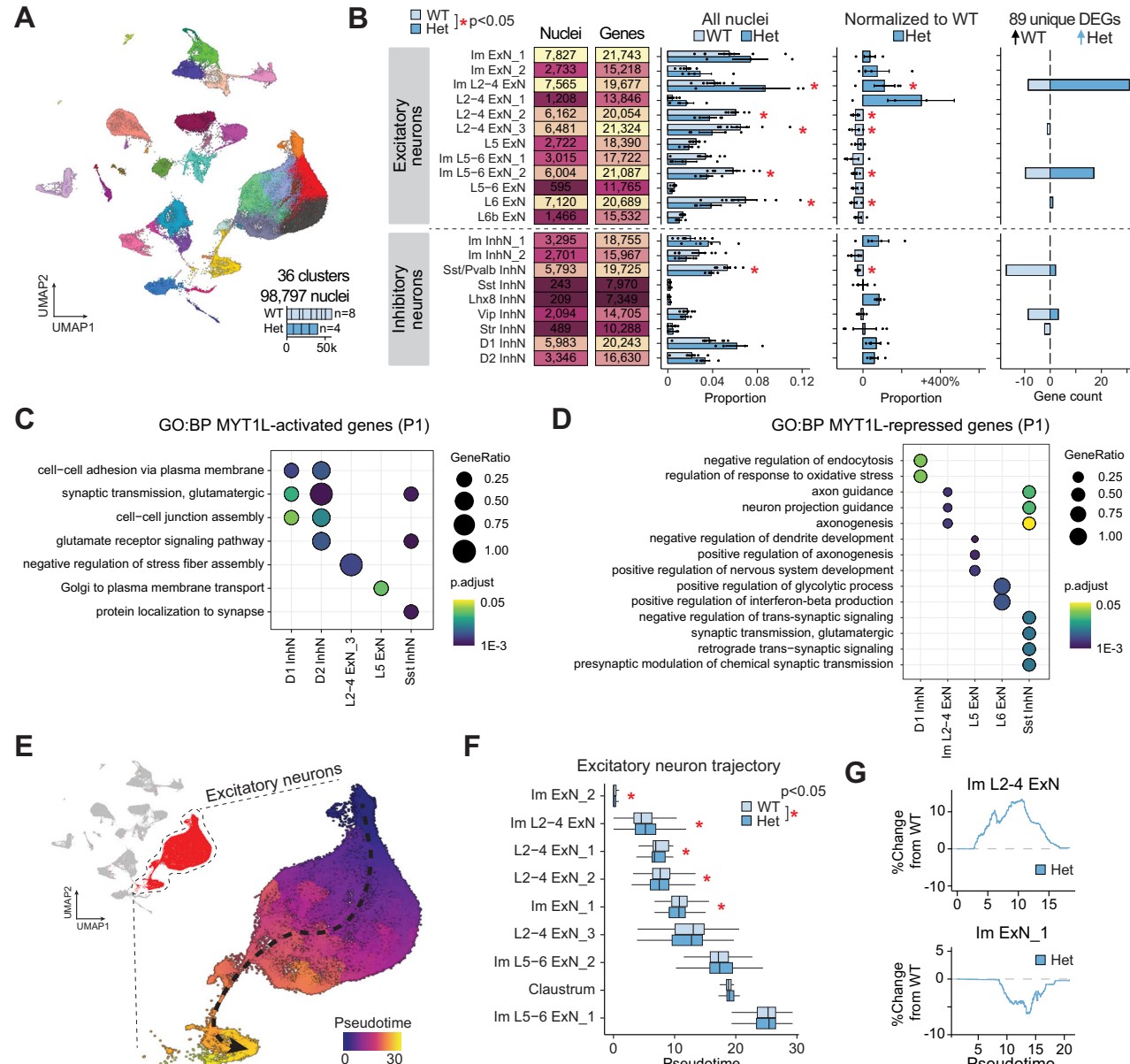

**Fig. 4 | Loss of MYT1L disrupts proportions of excitatory neurons at P1. A** UMAP projection showing 98,797 nuclei in 36 clusters from the forebrain of P1 MYT1L WT (n = 8) and Het (n = 4) animals. **B** From left to right: summary plot showing the numbers of nuclei and genes detected in each cluster; bar plot displaying the mean ± SEM relative proportions of nuclei in each annotated cell cluster for MYT1L WT and Het genotypes; mean ± SEM proportions of Het normalized to WT; and number of differentially expressed genes (DEGs) per cell type that are upregulated in WT (light blue; n = 8 biological replicates) and upregulated in Het (medium blue, n = 4 biological replicates) (*FDR adjusted p < 0.05, moderated t-test). Dot plots showing enriched GO biological processes terms in (**C**) MYT1L-activated (decreased expression in Het) and (**D**) MYT1L-repressed (increased expression in Het) DEGs. Overrepresentation of genes within GO terms was determined using a one-sided

hypergeometric test, with p values adjusted for multiple testing using the Benjamini-Hochberg method. **E** UMAP visualization of excitatory neuron clusters colored by pseudotime. The arrow indicates the inferred developmental trajectory from early (blue) to late (yellow) pseudotime. **F** Distribution of excitatory neuron subtypes along the pseudotime axis for WT (light blue; n = 8 biological replicates) and Het (blue; n = 4 biological replicates). Box plots show median (center line), interquartile range (box), and whiskers extending to 1.5 times the interquartile range. Asterisks indicate statistically significant differences between WT and Het distributions (*FDR adjusted p < 0.05, two-sided Kolmogorov-Smirnov test). **G** Percentage change in cell proportion along pseudotime in Het samples normalized to WT for two excitatory neuron subtypes: Im L2-4 ExN (top) and Im ExN_1 (bottom).

Intriguingly, we also observed a consistent increase in the proportion of DARPP32-positive striatal inhibitory neurons in MYT1L Het cortex at E14 (Fig. 1J), P1 (Fig. 4B), and P21 (Supplementary Fig. 5). This observation is unlikely to be an artifact of striatal tissue contamination during cortical dissections, given the consistent proportion difference between Het and WT samples across biological replicates at three time points. We hypothesized that MYT1L deficiency might result in the mislocalization of these cells during development, causing them to migrate to the cortex instead of their typical striatal destination. To validate this

finding and gain spatial information not captured by snRNAseq, we performed immunofluorescence staining to directly visualize and quantify the distribution of DARPP32 and NeuN double-positive neurons in situ from three different regions spanning the cortex (Fig. 5E, F). We confirmed the increase in DARPP32 and NeuN double-positive neurons in the deep layer of the cortex (Fig. 5G), validating the snRNAseq results and offering insight into the potential migration defects associated with MYT1L deficiency. Further investigation is needed to elucidate the mechanisms underlying this phenomenon.

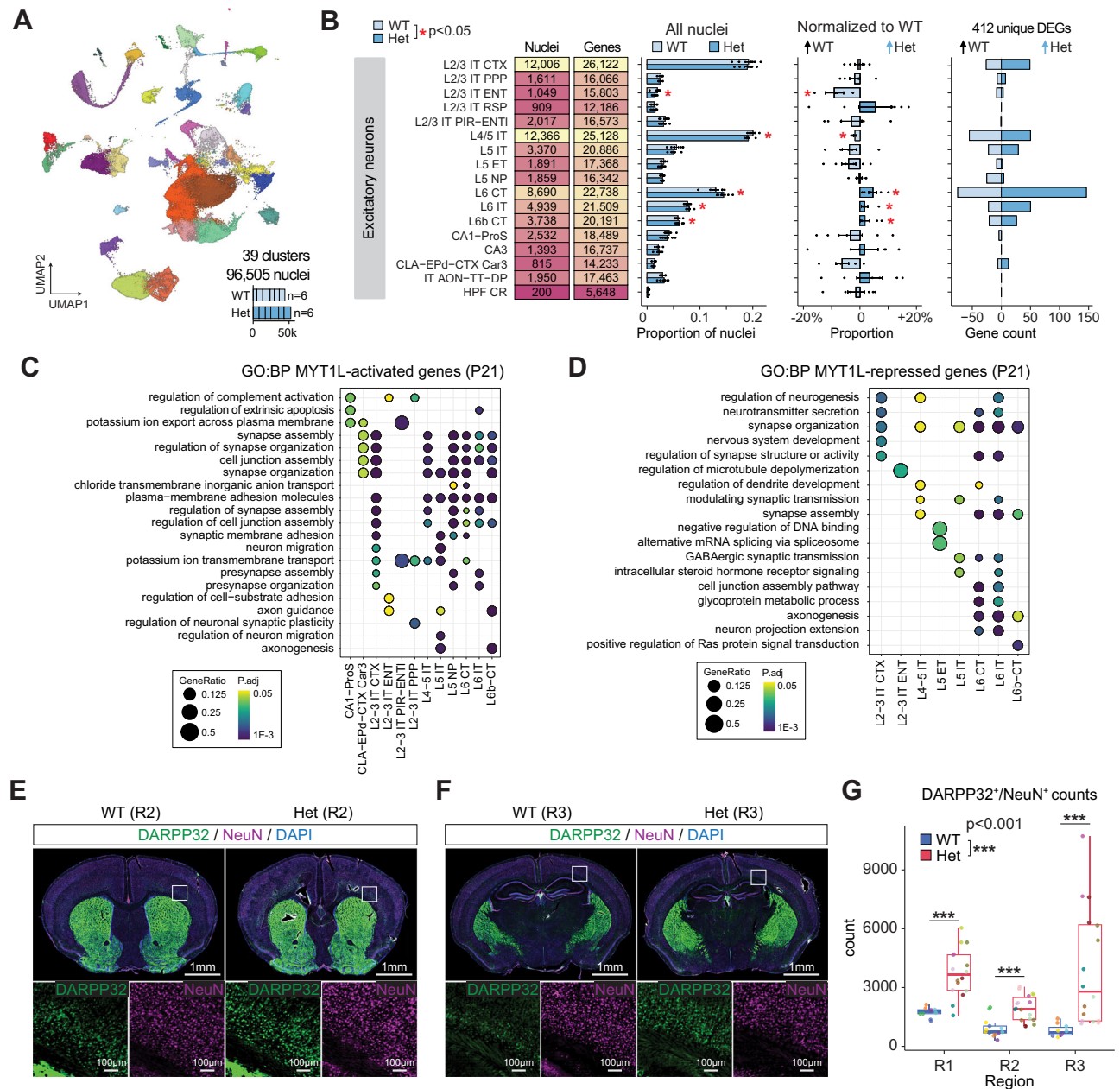

**Fig. 5 | Sensitivity of excitatory neurons persist throughout neurodevelopment to P21. A** UMAP projection showing 96,505 nuclei in 39 clusters from the cortex of P21 MYT1L WT (n = 6) and Het (n = 6) animals. **B** From left to right: summary plot showing the numbers of nuclei and genes detected in each cluster; bar plot displaying the mean ± SEM relative proportions of nuclei in each annotated cluster for MYT1L WT and Het genotypes; mean ± SEM proportions of Het normalized to WT; and number of differentially expressed genes (DEGs) per cell type upregulated in WT (light blue; n = 6 biological replicates) and upregulated in Het (blue; n = 6 biological replicates). Dot plots showing enriched GO biological processes terms in (**C**) MYT1L-activated (decreased expression in Het) and (**D**) MYT1L-repressed (increased expression in Het) DEGs. Overrepresentation of genes within GO terms was determined using a one-sided hypergeometric test, with p values adjusted for multiple testing using the Benjamini-Hochberg method. **E**, **F** Representative immunofluorescence images of brain sections from WT and MYT1L Het mice from a medial region (R2) and posterior (R3) region of the cortex. Upper panels show whole-brain sections stained for DARPP32 (green), NeuN (magenta), and DAPI (blue). Lower panels show higher magnification of the boxed areas in the upper panels, showing DARPP32 and NeuN staining separately. **G** Box plots showing quantification of DARPP32 and NeuN double-positive cell counts in three cortical regions (R1, R2, and R3) of WT and MYT1L Het mice. Each dot represents a counted section and is color-coded by animal. Box plots show median (center line), interquartile range (box), and whiskers extending to 1.5 times the interquartile range. Statistical significance was assessed using a linear mixed model comparison to test the effect of genotype while accounting for section differences and individual mouse variability. Significance is indicated by asterisks (***FDR adjusted p < 0.001, ANOVA).

## Disruptions in excitatory neurons are detectable across all timepoints

To deepen our understanding of the developmental progression from E14 progenitors to the end of neurogenesis at P1 to terminally differentiated cell types at P21, we integrated the three datasets together, analyzing a total of 412,132 nuclei. Integrated UMAP visualization revealed distinct clusters corresponding to major cell classes, including progenitors, excitatory and inhibitory neurons, glia, and other non-neuronal cell types. The 54 identified clusters showed clear developmental progression from E14 to P21, reflecting the maturation of various cell types over time (Fig. 6A). Analysis of cell class proportions across developmental stages confirmed the expected decrease in

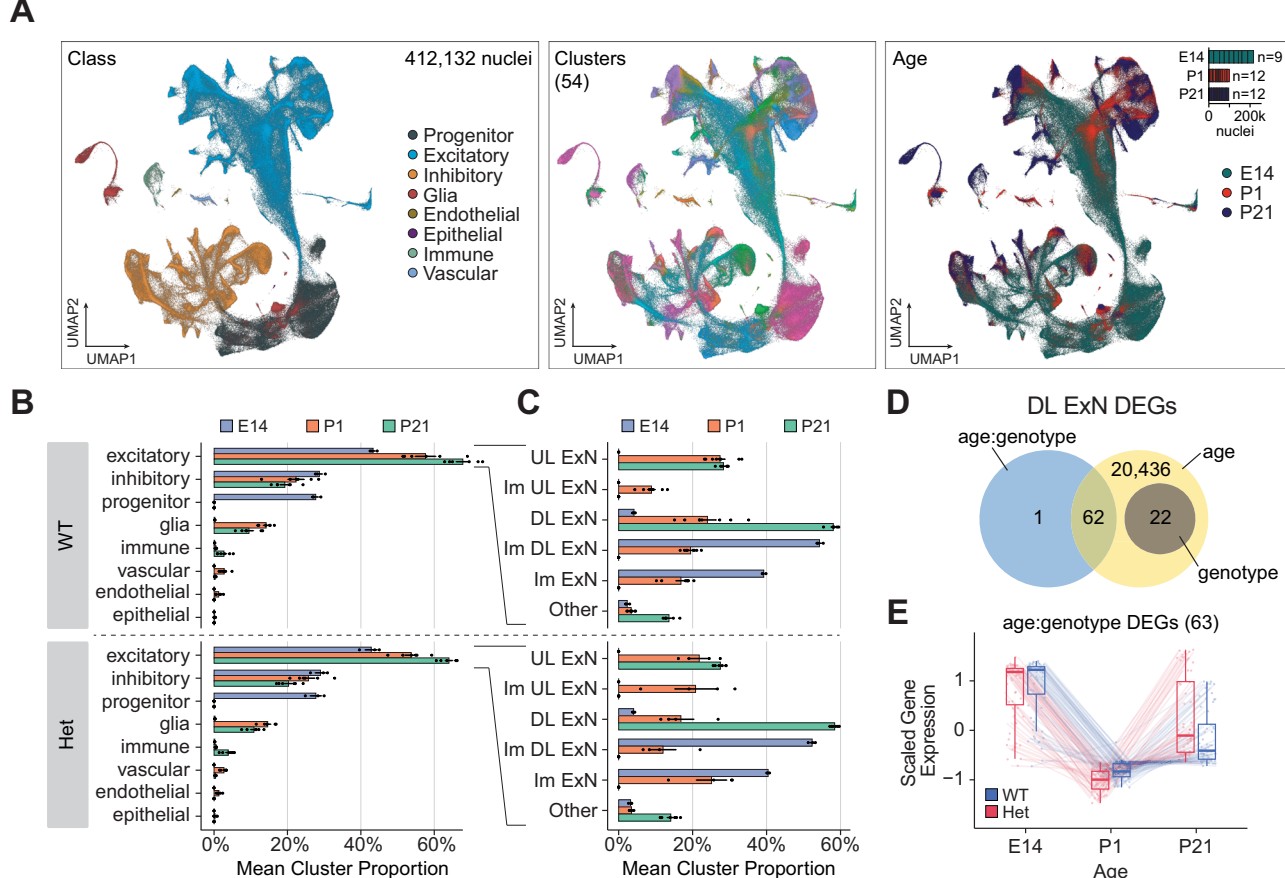

**Fig. 6 | Integrated analysis of MYT1L deficiency across neurodevelopment.**
**A** UMAP projection showing integrated data of 412,132 nuclei from E14, P1, and P21 datasets. Left panel: nuclei colored by cell classes. Middle panel: nuclei colored by 54 distinct clusters. Right panel: nuclei colored by developmental stage.
**B** Mean ± SEM proportions of major cell classes across developmental stages in WT (top; n = 17 animals) and Het (bottom; n = 13 animals). **C** Mean ± SEM proportions of excitatory neuron subtypes across developmental stages in WT (top; n = 17 animals) and Het (bottom; n = 13 animals). UL ExN: upper layer excitatory neurons, DL ExN:

deep layer excitatory neurons, Im: immature. **D** Venn diagram showing the number of differentially expressed genes identified by pseudobulk analysis in deep layer excitatory neurons (DL ExN) associated with age, genotype, and age:genotype interaction. **E** Expression patterns of genes showing significant age:genotype interaction across developmental stages in WT (blue) and Het (red) animals. Each line represents a gene, with box plots showing the distribution of scaled gene expression values. Box plots show median (center line), interquartile range (box), and whiskers extending to 1.5 times the interquartile range.

proportion of progenitor cells while excitatory and inhibitory neuron populations increased from E14 to P21, consistent with the progression of neurogenesis and maturation. (Fig. 6B). Focusing on the excitatory neurons, we observed a shift from immature to mature neurons across development in both genotypes. However, MYT1L Het animals showed subtle but consistent alterations in the proportions of deep layer (DL) and upper layer (UL) excitatory neurons (Fig. 6C).

To assess the transcriptional impact of MYT1L deficiency across development, we performed pseudobulk differential expression analysis on 80,866 deep layer excitatory neurons (DL ExN). We identified a total of 20,521 DEGs, with the vast majority associated with developmental stage (age) (Fig. 6D). Importantly, we found 63 DEGs showing an interaction between age and genotype. Further examination of the age:genotype interaction DEGs revealed distinct expression patterns across development for WT and Het animals (Fig. 6E). Some of these genes showed increased expression at P21 similar to E14 levels, such as NCOR2, SKI, and SMURF1, which are negative regulators of TGF-beta signaling[40,41]. This would likely result in significant suppression of TGF-beta signaling, potentially leading to disruptions in differentiation, maturation, and migration. Further investigation is needed to determine the extent and cellular context of the overexpression, as well as potential compensatory mechanisms activated in response to these changes.

The integrated analysis across multiple developmental timepoints reveals persistent and dynamic effects on cortical development. The alterations in cell type proportions, particularly in excitatory neuron subtypes, coupled with the mis-localization of DARPP32-positive neurons, highlight the critical role of MYT1L in proper neuronal migration and positioning. The transcriptional changes observed, especially those showing age-genotype interactions, underscore the complex and long-lasting impact of MYT1L deficiency on gene expression programs. The potential disruption of TGF-beta signaling suggests consequences for neuronal maturation. These findings not only elucidate the molecular and cellular mechanisms underlying MYT1L syndrome but also provide insights into the broader processes governing cortical development and the establishment of neuronal identity.

## Discussion

In this study, we analyzed the transcriptomes of 412,132 nuclei across neurodevelopment in a model of MYT1L mutation, delineating the molecular and cellular consequences of loss of MYT1L. The massively parallel barcoding technology enabled cost-effective integration of multiple biological replicates per genotype at each time point, minimizing false discoveries and artifacts associated with snRNAseq data sparsity. By leveraging single-cell atlases of the mouse brain as

references, we have advanced our understanding of how the disruption of a single transcription factor can perturb neurodevelopment and maturation processes.

Analysis of the developmental trajectory at E14, P1, and P21 led us to propose a unifying theory: MYT1L deficiency causes progenitors to first precociously differentiate, then mature slowly into functional neurons, with deep layer excitatory neurons being particularly susceptible. At E14, this slowed maturation results in fewer deep layer neurons compared to WT. MYT1L deficiency may lead to a premature fate switch in some progenitors due to altered expression of temporal identity genes that regulate the transition from generating deep to upper layer neurons. This may explain the continued decrease in deep layer neurons and increased proportion of upper layer neurons at P1. Intriguingly, at P21, we observe greater proportions of deep layer neurons in the MYT1L Hets, consistent with histology analysis[11], possibly due to a selective survival advantage during the postnatal wave of apoptosis. Further investigation into cell death patterns across cortical layers during development is needed.

Our results demonstrate that MYT1L deficiency causes a delay in neuronal maturation at E14 and P1, with dysregulation of maturation-related regulatory programs persisting through P21. The progressive increase of DEGs in Hets across these timepoints suggests that the consequences of MYT1L deficiency are sustained beyond the peak of MYT1L expression. This temporal progression can suggest two key implications: first, an amplification of indirect effects, where MYT1L's role as a transcriptional regulator could initiate a gradually expanding network of indirect effects on gene expression as development proceeds; and second, there may be a prolonged potential window for postnatal therapeutic interventions, given the continued increase in the effects of MYT1L deficiency after birth. Our findings reveal that MYT1L functions predominantly as a transcriptional repressor, modulating gene expression programs linked to key developmental processes such as axon guidance, neuron migration, and cell fate commitment. MYT1L-repressed pathways exhibited gene dose-responsiveness, while MYT1L-activated genes, primarily involved in synaptic function and neurotransmission, showed greater tolerance to haploinsufficiency. The dysregulated genes were enriched for TFs and epigenetic regulators, potentially triggering a cascade of downstream effects stemming from MYT1L perturbation. Using a brain region and age-matched MYT1L CUT&RUN dataset, we observed that most effects at E14 were indirect, with the percentage of direct effects increasing at P21. However, since MYT1L recruits the SIN3B deacetylation complex, many "indirect" regulatory targets identified by CUT&RUN may actually represent direct responses to earlier, unmeasured binding events, as deacetylated histones can maintain repressive epigenetic "memories"[42]. A deeper analysis of histone states could help disentangle this phenomenon. Nonetheless, these findings highlight the cell type-specific and developmental stage-specific nature of MYT1L's function, demonstrating its critical role in orchestrating neuronal maturation and gene expression during brain development.

To date, three transgenic mouse models disrupting different exons of MYT1L converge on a hyperactivity phenotype, while other behavioral outcomes vary, likely due to differences in assessment methods[12,13,15], or differences in the molecular consequence of each mutation. Weigel et al.[14] performed scRNAseq on the neonatal (P0) forebrain tissue from the mouse model described in Wohr et al.[15] Unlike our model, where MYT1L homozygous KOs are not viable postnatally and die at birth, their KO animals survive the first few days of postnatal development, allowing for P0 analysis. They observed a decreased number of newly formed neurons in the subventricular zone and increased expression of non-neuronal gene expression programs in KOs, potentially disturbing neuronal identity. Consistent with their findings, we noted a slight upregulation of mouse embryonic fibroblast (MEF) signature genes at P21 in Hets, albeit with a minor effect size. Both studies identified L5/6 neurons as having the greatest

number of DEGs. Interestingly, Weigel et al. also observed a moderate increase in the proportion of striatal inhibitory neurons in MYT1L Hets, aligning with our findings. A table summarizing results from this, and previous studies can be found in Supplementary Table 1. Additional experiments are needed to fully interpret these observations and their implications for cortical development in the context of MYT1L deficiency.

While MYT1L is a neuron-specific TF, it had been uncertain whether its loss may exert non-cell autonomous effects on surrounding glia during postnatal development. Our analysis revealed a relatively higher proportion of oligodendrocytes and microglia in P1 and P21 MYT1L Hets compared to WT, but we did not detect any DEGs in these cell types. However, with relatively low numbers of cells and gene counts in these clusters, there may be differences below our threshold to detect. These findings suggest the possibility of MYT1L-mediated effects on oligodendrocyte and microglia numbers, yet further investigations with increased cell numbers are needed to elucidate the nature and extent of these effects. What was abundantly clear in our data at both time points was the profound, cell type-specific transcriptional responses to MYT1L deficiency, especially in deep layer excitatory neurons.

A striking observation from our analysis of pseudobulk DEGs reveals that around 15% of these DEGs overlap with SFARI gene candidates and display significant dysregulation in deep layer excitatory neurons, particularly the L5-6 ExN_1, L5-6 ExN_2, and Im L6 ExN clusters (Supplementary Fig. 2). Interestingly, expression levels of these genes were elevated in KOs compared to WTs, hinting at the possible loss of a repressive mechanism. Despite the majority of these genes not being identified as direct targets in the E14 MYT1L CUT&RUN analysis, it is important to note that the CUT&RUN dataset only includes gene targets based on MYT1L occupancy in promoter regions, omitting potential targets influenced by distal regulatory elements, as it remains challenging to systematically link long-distance enhancers to specific gene targets. Nevertheless, the observed differential expression allows us to hypothesize that MYT1L may function as a transcriptional regulator, influencing SFARI gene expression directly or indirectly, or through mechanisms like epigenetic memory. Ultimately, the disrupted pathways we've identified represent a core set of pathways that are critical for proper neurodevelopment.

While our study provides valuable insights into the role of MYT1L in neurodevelopment, it is important to acknowledge its limitations. Massively parallel barcoding has significantly enhanced snRNAseq throughput, but even greater numbers of nuclei are needed to fully capture the complexity of cortical development. Our analysis of over 400,000 nuclei demonstrates specificity in the differences we have identified, but it may lack sensitivity due to limitations inherent to single-cell/nucleus technology. There may be additional cell types or subtle changes in cell type proportions that could only be detected by analyzing an even larger number of nuclei. This limitation in sensitivity is analogous to the challenge single-cell methods face in detecting differentially expressed genes. This is particularly crucial when examining the consequences of MYT1L, where the effect size of haploinsufficiency is subtle and varies across cell types and developmental stages. Our P1 dataset exemplifies this challenge, with relatively less statistical power due to an imbalanced sample size. Additionally, the comparison of WT to Het animals further reduce the effect size. In contrast, our E14 dataset, comparing WT to KO, contained nearly 2.2 times the number of nuclei with a greater effect size and consequently yielded more differentially expressed genes. The P21 dataset, while also comparing WT to Het, does not have the variation associated with developmental trajectories, and thus we were powered to detect smaller changes. This underscores the need for greater depth in single-cell studies of neurodevelopmental disorders to detect nuanced effects. Increased resolution could potentially reveal additional cell type-specific responses to MYT1L deficiency and provide a more

comprehensive understanding of its impact across all cortical layers and cell types. Future studies with even greater cellular depth and more biological replicates could further refine our findings and potentially uncover additional nuances in the developmental trajectories affected by MYT1L mutation.

In conclusion, our comprehensive analyses across developmental stages underscore the pivotal role of MYT1L in neuronal maturation and development. These findings reveal that the developmental trajectory and transcriptional landscape of excitatory neurons are markedly altered by MYT1L deficiency, with effects persisting from early neurogenesis through adolescence. This study not only advances our understanding of the genetic and molecular foundations of neuronal development but also demonstrates how we can deeply characterize genetic perturbations at scale to investigate the enduring impact of NDD associated mutations on the maturation and function of the brain.

## Methods

All animal studies were approved by and performed in accordance with the guidelines of the Animal Care and Use Committee of Washington University in Saint Louis, School of Medicine (Protocol 23-0138) and conformed to NIH guidelines for the care and use of laboratory animals.

### Animals and tissue collection

Animals were housed in controlled environments with a 12:12-h light-dark cycle, constant temperature (20–22 °C) and relative humidity (50%), and *ad libitum* access to standard laboratory diet and water. The C57BL/6-*Myt1l*[em1Jdd]/J (Myt1l S710fsX[12]; Jackson Laboratories 036428) line was maintained with breeding pairs consisting of a Myt1l Het and an in-house C57BL/6 J mouse. In-depth molecular and behavioral characterization of the mice is previously published[12]. The transgenic line was refreshed every 8–10 generations by backcrossing to freshly obtained C57BL/6 J males and females from Jackson Laboratories. Upon weaning at P21, the animals were group-housed by sex and genotype. To obtain homozygous embryos, timed pregnant Myt1l Het x Het breeding pairs were set up, with the first morning after finding a vaginal plug considered E0.5.

To harvest E14-14.5 embryos, the pregnant dams were deeply anesthetized with isoflurane followed by decapitation. Death was confirmed by absence of pulse, breathing, and response to firm toe pinch. The embryos were rapidly dissected in HBSS on ice and decapitated. The brains were extracted, meninges were removed, forebrains dissected, flash frozen in liquid nitrogen, and stored at −80 °C. Tail tissue was collected for gDNA isolation and genotyping. The forebrains from P1 pups and cortical tissue from P21 pups were similarly collected and stored at −80 °C. For the E14 cohort, mixed sexes from 3 biological replicates per WT, Het, and KO genotypes were used, totaling 9 samples. The P1 and P21 cohorts included only WT and Het animals, as KOs are not viable postnatally. The P1 cohort consisted of 12 animals, with 8 WT (5 male, 3 female) and 4 MYT1L Het (3 male, 1 female). The P21 cohort consisted of 12 animals: 3 male WT, 3 female WT, 3 male MYT1L Het, and 3 female MYT1L Het.

### Genotyping

Tissue (tail biopsy, ear punch, or toe clipping) was obtained from each animal and placed in a PCR tube. To confirm the genotype from processed nuclei samples, 2 µl nuclei suspension was added to a PCR tube. 100 µl lysis buffer (25 mM NaOH, 0.2 mM EDTA, pH 12) was added to each tube and incubated at 99 °C for 60 min in a thermocycler. Once the samples cooled to room temperature, 100 µl 40 mM Tris-HCl pH 5 was added to neutralize the alkaline lysis buffer. The crude lysate containing genomic DNA (gDNA) was stored at 4 °C. Three reactions were performed for each animal to genotype the MYT1L WT allele (For: 5′-ATGTCGCAGTAGCCAAGTC-3′, Rev: 5′-TCTTGCTACACGTGCTACT-

3′), MYT1L mutant allele (For: 5′-ATGTCGCAGTAGCCAAGTC-3′, Rev: 5′-TCTTGCTACACGTACTGGA-3′), and SRY (For: 5′-TTGTCTAGA-GAGCATGGAGGGCCATGTCAA-3′, Rev: 5′-CCACTCCTCTGTGA-CACTTTAGCCCTCCGA-3′) to determine sex. The PCR conditions for genotyping with allele specific PCR primer pairs involved mixing 1 µl of the crude gDNA with 5 µl Phusion High-Fidelity PCR Master Mix, 1 µl 10 µM MYT1L WT or mutant F/R primer mix, 1 µl 10 µM B-actin F/R primer mix (For: 5′-AGAGGGAAATCGTGCGTGAC-3′, Rev: 5′-CAA-TAGTGATGACCTGGCCGT-3′), and 2ul ddH2O. Thermocycling conditions were as follows: 98 °C for 3 min; 35 cycles of: 98 °C for 10 s, 61 °C for 20 s, 72 °C for 20 s; 72 °C for 5 min; and 4 °C hold. For SRY, 1 µl crude gDNA was added to a master mix containing 5ul OneTaq Quick-Load 2X Master Mix, 1 µl 10 µM SRY F/R primer mix, 1 µl 10 µM B-actin primer mix, and 2 µl ddH2O. Thermocycling conditions were as follows: 94 °C for 3 min; 35 cycles of: 94 °C for 10 s, 60 °C for 20 s, 68 °C for 20 s; 68 °C for 5 min; and 4 °C hold. Multiplexing B-actin not only confirms the presence of gDNA but also minimizes non-specific amplification of the MYT1L mutant band in WT samples. PCR products were run on a 1% agarose gel and visualized with GelRed.

### Injection of AAV-calling cards reagents

Calling Cards is a method to longitudinally record protein-DNA interactions over time in tissues[43–45]. The constructs hyPB and H2b-tdT-SRT (Addgene 203393) were packaged into AAV9 viral particles by the Hope Center Viral Vectors Core at Washington University. The titer was determined by qPCR and standardized to $1 \times 10^{13}$ vg/ml. A step-by-step protocol for transcranial injections is described in Yen et al.[45]. Briefly, the AAVs were mixed 1:1 and transcranially injected into the ventricles of P0-1 pups from MYT1L WT x Het breeding pairs. At P7, toe tissue was collected for genotyping. At P21, the animals were deeply anesthetized with isoflurane and perfused with ice cold DPBS. The brain was harvested, tdTomato fluorescence was verified using a handheld fluorescence flashlight (Nightsea Xite-GR), the cortex was dissected, and the tissue was flash frozen in liquid nitrogen and stored at −80 °C. The tissue was processed following the "nuclei isolation and fixation" section below.

### Nuclei isolation and fixation

In this study, nuclei from E14, P1, and P21 prefrontal cortices were isolated from flash frozen tissue. The samples for this study were randomly selected from available banked tissue across 3 litters of E14 animals, 4 litters of P1 animals, and 4 litters of P21 animals that have been grouped by genotype. Blinding was not used since we needed to determine which samples to process. The brain tissues were Dounce homogenized in ice-cold homogenization buffer (10 mM Tris-HCl pH 7.4, 10 mM NaCl, 3 mM MgCl$_2$, 1 mM DTT, 1X complete EDTA-free Protease Inhibitor (Roche 4693132001), and 0.2U/ul RNasin Inhibitor (Promega N2515)) using a 2 ml KIMBLE KONTES Dounce Tissue Grinder (DWK 885300-002) with 15 strokes with the "A" large clearance pestle, followed by 15 strokes of the "B" small clearance pestle. The homogenate was transferred to a 15 ml centrifuge tube. Walls of the homogenizer tubes were washed with 1 ml of homogenization buffer and combined with the homogenate in the 15 ml tube. The nuclei were pelleted by centrifugation in a swinging bucket rotor (Eppendorf S-4-104) at $500 \times g$ for 5 mins at 4 °C. The supernatant was aspirated and discarded.

For E14 and P1 samples, the pellets were washed twice with 1 ml nuclei wash buffer (DPBS, 1% BSA, and 0.2 U/µl RNase inhibitor), filtered through a 40 µm Flowmi cell strainer (Sigma-Aldrich BAH136800040), and manually counted using a hemocytometer with Trypan Blue or propidium iodide (Biotium 40017).

For P21 samples, the pellets after the first centrifugation were resuspended in 1 ml homogenization buffer. Density gradient centrifugation was performed to purify the nuclei from cellular debris and myelin generated during tissue dissociation. A 50% iodixanol-PBS working solution was prepared by diluting the stock solution of 60% w/v Optiprep (Sigma-Aldrich D1556) with DPBS (ThermoFisher

14190136). A 35% (w/v) iodixanol solution was made by diluting the 50% iodixanol with DPBS. To make the 25% Iodixanol layer, 1 ml 50% iodixanol was added to 1 ml of the homogenate containing the nuclei and debris. This was carefully layered on top of 2 ml 35% iodixanol in a clear polycarbonate tube (Beckman Coulter 355672) and centrifuged using an Allegra 64R (Beckman Coulter 367586) at $10,000 \times g$ for 30 min at 4 °C in a S0410 swinging bucket rotor (Beckman Coulter 364660) with no braking. After the centrifugation, myelin, and cellular debris remaining at the top of the 25% iodixanol layer were aspirated and discarded. The purified nuclei at the interface of the 25% and 35% iodixanol layers were collected using a low retention P1000 pipette and transferred to a clean 15 ml centrifuge tube. The volume was brought up to 6 ml with nuclei wash and resuspension buffer and pelleted by centrifuging at $500 \times g$ for 5 min at 4 °C. The supernatant was carefully removed, washed once with nuclei wash buffer to ensure removal of carryover iodixanol, filtered through a 40 μm Flowmi cell strainer, and manually counted using a hemocytometer with Trypan Blue or propidium iodide.

The ScaleBio Sample Fixation Kit (Scale Biosciences 2020001) was used to fix the nuclei. For E14 and P21 samples, 500 k–2.5 M nuclei were resuspended in 500 μl calcium and magnesium-free DPBS and used as input according to the manufacturer's standard fixation protocol. For P1 samples, 100–300 k nuclei were resuspended in 50 μl calcium and magnesium-free DPBS and used as input according to the manufacturer's low sample input protocol. After fixation, the nuclei were manually counted once more and checked for quality using a microscope with a 60× objective. The nuclei were then stored at −80 °C until all samples have been collected and fixed.

### Single-nucleus RNAseq library preparation and sequencing

Libraries were prepared from fixed E14, P1, and P21 nuclei separately. For the E14 cohort, a total of 9 samples (3 biological replicates of a mix of males and females per MYT1L WT, Het, and KO genotypes) were used. For the P1 cohort, 12 samples consisting of 8 biological replicates for MYT1L WT and 4 biological replicates for MYT1L Het genotypes were used. For the P21 cohort, 12 samples consisting of 3 biological replicates per sex per MYT1L WT and Het genotypes were used. The day of the library preparation, the frozen fixed nuclei were thawed on ice and each sample was counted twice using a hemocytometer.

For the E14 and P1 samples, the ScaleBio Single Cell RNA Sequencing Kit v1.0 (Scale Biosciences 2020008) was used according to manufacturer's instructions. Nuclei from each sample were loaded at 10,000 nuclei per well to the 96-well Indexed RT Oligo Plate to add the RT barcode and UMI onto each transcript during reverse transcription. By loading each sample into a distinct set of wells, the RT barcodes can serve as sample identifiers, enabling all genotypes to be processed on the same plate in a single batch per age. The nuclei from each well were then collected and pooled using the Scale Biosciences' supplied collection funnel, mixed, and distributed across the 384-well Indexed Ligation Oligo Plate where the Ligation Barcode was added to each UMI-RT barcoded transcript. Then, the nuclei were once again collected and pooled using another collection funnel and counted with a hemocytometer with Trypan Blue. A total of 1600 nuclei were distributed per well of the 96-well Final Distribution Plate. In each well, second strand synthesis was performed followed by a cleanup step. The PCR products were then tagmented followed by an indexing PCR step to add a third barcode to each well. 5ul from each of the 96 libraries were pooled and cleaned using 0.8X SPRIselect beads (Beckman Coulter B23317). The average fragment size of the final library was quantified using a High Sensitivity D5000 Screentape (Agilent). The library concentration was quantified using the NEBNext Library Quant Kit for Illumina (New England Biolands E7630S). The libraries were sequenced on a shared S4 flowcell on a NovaSeq6000 (Illumina) instrument or a shared 25B flowcell on a Novaseq X Plus (Illumina) to a target depth of 10,000 reads per nucleus.

For the P21 samples, the protocol described above was followed through the cleanup step. Prior to tagmentation, 3 μl (half of the total volume) was transferred to a clean 96-well PCR plate to create a Calling Cards Final Distribution Plate. This plate was set aside to pilot single-nucleus Calling Cards (snCC) library preparation, the results of which will be reported in a future methods paper. The remaining 3ul was used for the remainder of the ScaleBio protocol with slight modifications. To account for the reduced volume of template input, the volumes for all subsequent steps have been halved to keep all reaction proportions the same. Additionally, the Indexing PCR program was increased to 16 cycles instead of 14. The libraries were pooled, cleaned, and quantified as described above according to manufacturer's instructions. This library pool was sequenced on a shared 25B flowcell on a Novaseq X Plus (Illumina) instrument to a target depth of 10,000 reads per nucleus.

### snRNAseq data processing

Base calls were converted to fastq format and demultiplexed by Index1 barcode by the Genome Technology Access Center at the McDonnell Genome Institute (GTAC@MGI). Combinatorial barcode demultiplexing, barcode processing, adapter trimming, read mapping to the mm10 reference genome, single-nuclei counting, and generation of the feature-barcode matrices were done using ScaleRna v1.4 (https://github.com/ScaleBio/ScaleRna). CellFinder, an EmptyDrops-like cell calling method from ScaleRna v1.5 was applied to account for ambient RNA and recover nuclei with low total RNA content. The filtered count matrices were brought into Seurat for downstream analyses of the E14, P1, and P21 groups separately.

For quality control, multiplets were removed by DoubletFinder[46], followed by a UMI-gene cutoff of 800–6000 UMIs and 300–3000 genes for each sample. Barcodes with % mitochondrial reads >1 and $\frac{\log(genes)}{\log(UMIs)} < 0.9$ were removed. After quality control filtering, the E14 dataset contains 216,830 nuclei across all samples, with a median of 3204 UMIs and 1819 genes per nucleus. The P1 dataset contains 98,797 nuclei with a median of 1963 UMIs and 1176 genes per nucleus. The P21 dataset contains 96,505 nuclei with a median of 3447 UMIs and 1597 genes per nucleus. The count matrices were log2 normalized, centered, and scaled using a scaling factor of 10,000. The top 3000 most variable genes were identified using dispersion and mean expression thresholds. Principal component analysis (PCA) was then performed on the top 100 components followed by dimensionality reduction by UMAP. We used an iterative unsupervised clustering approach using the Louvain algorithm. First, we clustered nuclei into the main cell classes (e.g., progenitors, excitatory neurons, inhibitory neurons, glia, and other). Then for each class, we performed another round of clustering to identify the cell types. The cluster resolution was optimized by testing a range of values (0–1.4 in steps of 0.2) and plotted into a clustering tree using clustree[47]. Cluster marker genes were defined by grouping the clusters, setting logfc.threshold = 0.5 and min.pct = 0.25, and comparing the fold changes between pct.1 and pct.2 using FindMarkers and the Wilcoxon rank-sum test. It is important to keep in mind that the Wilcoxon rank-sum test is suitable for this purpose, however, the p values can be inflated and should not be automatically considered as differentially expressed genes[29,30,48]. The identified marker genes were then referenced with marker genes from well-annotated datasets and the ABC atlas[19]. For the P21 dataset, the raw count data was saved in the h5ad format and the hierarchical mapping algorithm from MapMyCells (RRID: SCR_024672) was used to align the nuclei with the 10× Whole Mouse Brain (CCN20230722) reference. The mapping results were imported into R and annotations were transferred into the metadata.

### Differential gene expression analysis

To identify genes differentially expressed in MYT1L mutants compared to WT controls in each cell type, we created pseudobulk samples by summing single-cell expression counts for each gene within each cluster and

sample combination (e.g., RG_1 x E14-WT-1, RG_1 x E14-WT-2, RG_2 x E14-WT-1, etc.). We then used these count matrices and associated sample-level metadata with DESeq2 for differential analysis[29–31]. To assess whether the transcriptional response to MYT1L perturbation varied by sex, we initially tested a design (-sex + genotype + sex:genotype) that modeled both main effects of sex and genotype and their interaction. We found no significant sex × genotype interactions for genes on autosomes, with differential expression between sexes limited to genes on sex chromosomes. This likely reflects our limited statistical power to detect sex-specific effects. Given these findings, we proceeded with the more parsimonious genotype-only model for subsequent analyses. For each cluster, we created a DESeq2 object using the *DESeqDataSetFromMatrix* function with the design formula -genotype. Count normalization was performed using the median of ratios method, and the data were transformed for visualization using a regularized log transform. Principal component analysis and hierarchical clustering ensured sample-level quality control. No outliers were detected, and no samples or factors were removed. The differential expression analysis followed the standard DESeq2 workflow: normalizing library depth with a size factor, estimating gene-wise dispersion, refining the dispersion estimates using the apeglm method[49], and fitting a negative binomial model for each gene. Pairwise comparison used the Wald test, while the Likelihood Ratio Test evaluated differential gene expression changes across MYT1L gene doses by coding the number of functional alleles as ordinal factors (0 = KO, 1 = Het, and 2 = WT). Genes with a log2 fold change of at least 0.2 and FDR < 0.05 were identified as differentially expressed. This analysis was iteratively run for all cell types.

To calculate the DEG burden per cell type, we downsampled the data by randomly selecting 700 nuclei from each cell type before performing the differential gene expression analysis. This process was repeated 10 times, and the average number of DEGs from these normalized nuclei counts represents the degree of gene dysregulation per cluster. OPC and MG clusters were reported as "N/A" due to insufficient nuclei numbers for downsampling.

### Gene ontology (GO) enrichment analysis

Over-representation analysis of the differentially expressed genes was performed using the clusterProfiler package[50]. All expressed genes within a given cluster was used as background and GO Biological Processes ontology was used. A one-sided Fisher's exact test was used, and hypergeometric p values were FDR-corrected using Benjamini-Hochberg procedures. GO terms with an FDR > 0.05 were considered and redundant terms with an information content > 0.7 were removed using the *simplify* function.

### Regulatory network inference and analysis

Single-cell Regulatory Network Inference and Clustering (SCENIC)[32,51] was used to reconstruct regulon activity in single nuclei for the E14 dataset. Regulons, defined as transcription factors and their target genes, were identified through a multi-step process. First, we performed gene regulatory network inference using the GRNBoost2 algorithm, inputting a loom file containing the raw gene expression counts matrix and a list of mouse transcription factors to generate an adjacencies matrix representing co-expression modules. Next, we identified candidate regulons based on transcription factors-target gene interactions and refined these using cisTarget for motif discovery to eliminate potential indirect targets. Cell type-specific regulon activity was then computed using AUCell using an AUC threshold of 0.05 which assesses whether the genes in the signature are within the top 5% of expressed genes. The resulting AUC matrix, representing "regulon activity scores" for each transcription factor for each cell, was integrated into the E14 scanpy AnnData object. Finally, we subset the data by genotype and computed regulon specific scores (RSS) for each genotype. This approach enables us to compare transcription factor activities and cross-reference the differentially expressed genes across the genotypes at E14.

### Pseudotime and trajectory analysis

Pseudotemporal ordering of the E14 and P1 datasets was performed independently using Monocle3[37,52]. For E14, Cajal-Retzius, MG, and OPC clusters were excluded from the analysis to focus on the excitatory and inhibitory trajectories. For P1, only excitatory neurons were analyzed. The gene-count matrix was normalized by log+pseudocount and size factor to account for sequencing depth differences followed by scaling. A lower dimensional intermediate was calculated using the top 100 principal components from the top 5000 variable genes. Dimensionality reduction was performed using the *reduceDimension* function with the following parameters: (n_neighbors = 50, min_dist = 0.1, metric = "cosine"). The nuclei were then clustered and partitioned using the *cluster_cells* function. A principal graph was fit to the data and the cells were projected onto the graph using *learn_graph* with partitions. The earliest principal points within the excitatory and inhibitory trajectories were manually selected and independently assigned as the root state for the pseudotime computations.

To assess the effect of MYT1L loss on the timing of gene expression, the *graph_test* function with neighbor_graph = "principal_graph" parameter was used to identify differentially expressed genes along the trajectory. Genes with a q value < 0.05 were classified as pseudotemporal dynamic genes. The dataset was then subset by genotype to analyze the distributions of nuclei and expression of pseudotemporal genes across pseudotime. To obtain the temporal expression profile of TFs in excitatory neurons, we intersected the gene list from the WT excitatory trajectory with a list of mouse transcription factors curated from the AnimalTFDB 4.0 database[53]. The Kullback-Leibler divergence (KL Div) metric was used to measure the difference between the pseudotemporal TF expression distributions and compared to a null distribution generated by randomly shuffling the genotype labels.

### Brain sectioning and immunofluorescence staining

Brain tissues from postnatal day 21 (P21) mice were harvested after transcardial perfusion with ice-cold PBS followed by 4% (w/v) paraformaldehyde. The tissues were cryoprotected in graded sucrose solutions (15% and 30% w/v) and embedded in OCT compound. Frozen brains were sectioned on a cryostat into 70 μm free-floating coronal sections and stored in 1X PBS with 0.05% sodium azide at 4 °C.

For immunofluorescence staining, three sections from the anterior, medial, and posterior regions of the cortex were matched and selected from WT (n = 9) and MYT1L Het (n = 9) animals, totaling 54 sections. Tissues were washed three times with PBS (5 min each), then permeabilized with 0.1% (v/v) Triton X-100 in PBS for 15 min at room temperature. Sections were then incubated in blocking buffer 0.1% (v/v) Triton X-100 and 5% (v/v) normal donkey serum in PBS for 1 h at room temperature. Primary antibody incubation was performed overnight at 4 °C using the following antibodies diluted in blocking buffer: rabbit anti-DARPP-32 (1:200, Invitrogen MA5-32113) and mouse anti-NeuN (1:400, Invitrogen MA5-33103). The next day, tissues were washed five times in PBS (5 min each) and incubated for 1 h at room temperature with the following secondary antibodies diluted 1:400 in blocking buffer: donkey anti-rabbit Alexa Fluor488 (Invitrogen A-21206) and donkey anti-mouse Alexa Fluor568 (Invitrogen A10037). After two more PBS washes (5 min each), nuclear counterstaining was performed using 1 μg/mL DAPI in PBS for 15 min. Sections were washed once more before mounting on glass slides with ProLong Gold Anti-fade Mountant (Thermo Fisher Scientific P36930) and coverslipped. The edges were sealed with nail polish.

### Image processing and analysis

Immunofluorescence images were captured using a Zeiss AxioScan Z1 slide scanner equipped with a 10× objective lens, fluorescence filters for DAPI, Alexa Fluor488, Alexa Fluor568, and Alexa Fluor647, an Orca Flash sCMOS camera (Hammatsu C13440-20CU), and a mercury illumination source (Zeiss HXP120). A center of gravity focusing strategy

ensured precise stitching of the tissue sections. Tissue margins were autodetected at low magnification with a 5× objective, and exposure times were empirically optimized for each fluorophore to maximize signal intensity while minimizing photobleaching and background noise.

Image processing and quantification were conducted by an investigator blinded to animal genotype. Using ImageJ software, raw TIF images underwent background subtraction (rolling ball radius = 50 pixels) and unsharp masking (radius = 3 pixels, mask weight = 0.4). Binary masks for DARPP32 and NeuN channels were created using the "Make Binary" function, followed by pixel value normalization based on threshold settings. The "Image Calculator" function was used to multiply these masks with the original fluorescence images, identifying DARPP32 and NeuN double-positive regions of interest (ROIs). Cell counts were obtained using the "Analyze Particles" function. Once all images were identically processed, the investigator was unblinded and the statistical significance of differences by genotype was assessed using a linear mixed model comparison using the R package lme4. The full model (count ~ genotype + section + (1 | mouse)) was compared to a reduced model (count ~ 1 + section + (1 | mouse)) using ANOVA to test the effect of genotype while accounting for section differences and individual mouse variability.

## Statistics

No statistical methods were used to predetermine sample sizes. No animals were excluded from any analyses. Samples were generally littermates and genotypes were assigned randomly by the sperm at conception, with no input from investigators. For snRNAseq experiments, investigators were not blinded to the genotypes, however, all samples were processed in parallel on the same plates, batched by age. Detailed descriptions of statistical analyses for specific experiments are provided in their respective sections above.

## Reporting summary

Further information on research design is available in the Nature Portfolio Reporting Summary linked to this article.

## Data availability

The raw and processed data generated in this study have been deposited in the Gene Expression Omnibus database under the accession code SuperSeries GSE262368 and has also been deposited in the Neuroscience Multi-omic Data Archive (NeMO). The E14 MYT1L CUT&RUN dataset was downloaded from Gene Expression Omnibus under the accession code GSE222072. Source data with relevant raw data for each figure are provided with this paper. Source data are provided with this paper.

## Code availability

The ScaleRna Nextflow pipeline for processing raw Fastq reads to feature-barcode matrices is available at Github (https://github.com/ScaleBio/ScaleRna). The code needed to reproduce the key findings of this paper is found at Bitbucket (https://bitbucket.org/jdlabteam/yen-et-al-myt1l-snrnaseq/src/main/).

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

## Acknowledgements

We thank members of the Dougherty and Mitra laboratories for helpful discussions and feedback, particularly M. Vasek for critical reading and copyediting of the manuscript; Y. Li for helpful analysis suggestions; the DNA Sequencing and Innovation Lab (DSIL) and the Genome Technology Access Center at the McDonnell Genome Institute (GTAC@MGI) for their sequencing expertise and services; and F. Schlesinger and B. Biddy for bioinformatic support for the ScaleBio pipeline. This work was funded by grants from the National Institute of Mental Health (RF1MH117070, RF1MH126723, and R01MH124808 to J.D.D. and R.D.M.). A.Y. was supported in part by the National Human Genome Research Institute (T32HG000045). S.S. was supported in part by the Autism Science Foundation (22-007). Research reported in this publication was supported by the Eunice Kennedy Shriver National Institute of Child Health & Human Development of the National Institutes of Health under Award Number P50 HD103525 to the Intellectual and Developmental Disabilities Research Center at Washington University. Imaging was performed in part through the use of Washington University Center for Cellular Imaging (WUCCI) supported by Washington University School of Medicine, The Children's Discovery Institute of Washington University and St. Louis Children's Hospital (CDI-CORE-2015-505 and CDI-CORE-2019-813) and the Foundation for Barnes-Jewish Hospital (3770 and 4642). The funders had no role in conceptualization, study design, data collection, analysis, decision to publish, or preparation of the manuscript. The content is solely the responsibility of the authors and does not necessarily represent the official views of the National Institutes of Health.

## Author contributions

Project conceptualization: A.Y., R.D.M., and J.D.D. Method development, experiments, and data collection: A.Y., S.S., X.C., D.D.S., F.L., M.C., J.H-L., J.C., Z.A.L., and K.K.N. Formal analysis: A.Y., S.S., Y.W., Z.A.L., R.D.M., and J.D.D. Figures and data visualization: A.Y., S.S., Z.A.L., R.D.M., and J.D.D. Writing-original draft: A.Y. and J.D.D. Writing-review and editing: A.Y., S.S., J.C., R.D.M., and J.D.D. Project coordination: A.Y., R.D.M., and J.D.D. Funding acquisition: R.D.M. and J.D.D.

## Competing interests

D.D.S. and F.L. are employees of Scale Biosciences. The other authors declare no competing interests.
