## [Transparent Peer Review file · Nature Communications]

MYT1L deficiency impairs excitatory neuron trajectory during cortical development

Corresponding Author: Dr Joseph Dougherty

Version 0:

Reviewer comments:

Reviewer #1

(Remarks to the Author)

This manuscript by Yen and colleagues studies the genomic impact of loss of MYT1L on brain development at single cell resolution in the mouse brain. MYT1L is a transcription factor associated with NDD and prior studies had examined how loss impacts mouse behavior as well as gene expression in bulk RNA-seq. Here, the authors use snRNA-seq at two time points to characterize cell type specific gene expression changes. The authors also integrate their prior CUT&RUN data to determine the direct targets of MYT1L.

Overall, these experiments are important next steps to determine more finely the specific role of this transcription factor in brain development. The use of two time points is important as cell type proliferation and differentiation is dynamic over development.

Specific comments:

- 1) What is the point of Figure 1G? Not informative and not called out in the text.
- 2) For differential expression, the authors mention "aggregated" clusters. Did the authors recluster specific cell types, clean them up, and then run analysis or simply take the information from the cluster assignments from the original assignment at the all cells level? More details in the methods especially with respect to quality control could be described.
- 3) There obviously is no "right" way to perform differential gene expression analysis but the authors use a fairly conservative adjusted p-value of <0.1 . This leads to thousands of genes being called different. I would prefer a slightly more stringent cut-off of 0.05, which is a bit more standard, and see how the lists of genes change. I suspect there will still be hundreds of genes changing that make sense with the biological changes observed.
- 4) At E14, the authors mention that the lower layer neurons are the most affected by loss of MYT1L but this might be expected as upper layer neurons are still in the process of being generated and migrating. Moreover, the results at P21 show an increase in the number of lower layer neurons in contrast to the decrease observed at E14. How do the authors reconcile these results?
- 5) I'm not sure I understand the rationale to combine the E14 and P21 data. Those time points are very different in terms of brain development. I think it is better to compare the results of each time point analyzed separately.
- 6) The authors selected the two timepoints primarily because of the high expression of MYT1L in post-mitotic neurons. The differences between the experimental group and the control group are already striking at the E14 timepoint and continue through P21. The authors also show that the repressor activity of this gene is more prominent and that this effect is dose-dependent. A concern could be that the timepoints are overlooking critical changes at earlier stages of development. This is especially important as the manuscript's introduction states that understanding the timeline could open possibilities for interventions. Such a timeline remains unanswered by these time points, so I suggest the authors describe additional future experiments to further narrow the developmental window.
- 7) The functional connectivity changes are a bit hard to interpret because the within region connectivity was also reduced. Are these results due to more neurons or some other consequence of more neurons affecting connectivity? How do these changes ultimately affect the behavior of the mouse (the authors can speculate)? Are there effects on white matter tracts due to increased numbers of neurons?

Reviewer #2

(Remarks to the Author)

The authors are investigating the effect of Myt1l haploinsufficiency or knock-out on different brain cells at different times at single cell resolution. They first show that changes in the doses of Myt1l affects the proportion of different neurons at E14.5, followed by analysis of the transcriptional disruption in excitatory neurons due to loss of Myt1l. They concluded that there is a more immature transcriptional signature upon loss of Myt1l and that the sensitivity of excitatory neurons persists to postnatal development at least till P21. Furthermore, they show that Myt1l acts primarily as a transcriptional repressor during development.

Major revisions:

At E14.5 layer 6 neurons are the most differentiated and the rest of the postmitotic neurons are just out of intermediate progenitor phase and have not differentiated much, so how can you say layer 6 neurons are the most affected? It would be more interesting to look at P0 as this is the last time point to get KO mice and see the differences with mostly differentiated neurons. Especially as you see more layer 6 neurons at P21 and less upper layers.

Stain for the neurons that have different proportions to determine if what you see in the snRNA-Seq is also visible in the cortex, especially as the analysis is not in line with your histological analysis in Chen et al. 2021. How do you explain that? In line with that, Cell cycle disruptions have been shown in Myt1l mice, is there a difference between WT/HET/KO nuclei and their percentage in the cell cycle at E14.5?

Analysis of TF networks using SCENIC would be an interesting add on to get a better sense of the network of Myt1l

There is no analysis on inhibitory neurons but it is implied that the effect is stronger in excitatory neurons, please either perform the analysis and share it or rephrase it

Prior bulk RNAseq of the E14 MYT1L Het mouse cortex revealed an immature transcriptional signature when compared to WT. The use of immature in this context seems misleading, in Chen et al. it is discussed that there is earlier neuronal differentiation, which for me suggest faster development than in WT. Please clarify.

What are the pathways/GO term enrichments in between WT and HET mice at E14.5? Supplemental tables for DEGs and GO term analysis would be much appreciated.

Please give examples of the genes that are differentially expressed between WT and HET at P21.

Minor revisions

Myt1l het mice have microcephaly, did you account for that in your analysis at P21?

I am not entirely sure what this sentence means, please clarify: There were only 11 genes that were not exclusively up or downregulated across all cell types.

Reviewer #3

(Remarks to the Author)

In this manuscript, the Dougherty lab continues their examination of MYT1L deficient mice, this time providing a robust single cell genomics analysis of the developmental effects of Myt1l abrogation (WT, KO, and Het animals). In general, the results are compelling and the methods development with respect to the calling cards technique is interesting and potentially very valuable although it is only cursorily described. Although their findings are robust and important for better understanding of cortical neuron development, there are also some concerns that need to be addressed in a revised version of the paper.

Major concerns:

1. Related to fig. 1F, it is important to not only show the expression of Myt1l in all cells, but in each genotype separately as well. It is to be expected that the Myt1l knockout (KO) genotype has no expression signal, which serves as a crucial confirmation of the intended mutation.
2. The use of RNA velocity in a single nuclei RNAseq dataset is questionable. Quoting a recent review by the original authors of RNA velocity: "the assumptions such as constant degradation and nuclear export have not been conclusively verified [for single nuclei RNAseq datasets]" (Bergen, V., Soldatov, R. A., Kharchenko, P. V., & Theis, F. J. (2021). RNA velocity—current challenges and future perspectives. *Molecular Systems Biology*, 17(8), 10282. <https://doi.org/10.15252/MSB.202110282>). If the authors wish to apply this technique, they should demonstrate that their dataset met the prerequisites laid out in this authoritative paper.
3. Related to fig. 2C, it is unclear why the authors only performed differential expression (DE) analysis between wildtype (WT) and KO genotypes but did not directly compare WT and heterozygous (Het) genotypes.
4. The methods section is not complete and detailed enough. There is no description of the enrichment analysis procedure. Pseudotime and RNA velocity analyses were not described sufficiently.
5. Related to fig. 4B, since the authors only performed the single cell experiment workflow once, it is uncertain whether the differences in numbers of each cell type between genotypes accurately reflects real biological changes or simply a result of random sampling.

Minor concerns:

1. Related to fig. 2A: To identify transcriptional changes due to MYT1L deficiency, the authors first aggregated single nuclei data into pseudobulk samples, and then used a method (DESeq2) that is not designed specifically for single cell analysis. The reason for DE analysis after aggregation is unclear, given that DE techniques tailored for single cell datasets without aggregation are readily available.
2. Related to fig. 2A and supplemental fig. 1, when the authors tested overlap between DEGs and SFARI genes, they performed permutation by randomly selecting 932 genes, which is the number of SFARI genes. However, a more conclusive result would arise from sampling 1174 genes from all expressed genes and examine the overlap with SFARI genes each time. Although this may not affect the conclusion, the test needs to be set up reasonably.
3. There is at least one grand statement that may need some qualification. For example on page 4 (I think because there are no page numbers): "This suggests that there are no signals from the differentiating neurons that robustly influence the transcriptional identity of the proliferating progenitor pool at E14." This statement may be most pertinent to L6 neurons (and

not all differentiating neurons) and may be affected by the authors' definition of "robustly" and "transcriptional identity."

Version 1:

Reviewer comments:

Reviewer #1

(Remarks to the Author)

The authors have addressed all of my concerns and I have no further comments.

Reviewer #2

(Remarks to the Author)

All of my concerns have been addressed in the revision.

Reviewer #3

(Remarks to the Author)

I don't have any further concerns for this manuscript. The authors did a great job answering my questions.

Dear editor/reviewers,

Thank you for your thorough review of our manuscript. We appreciate your constructive feedback, which has helped improve the quality and clarity of our work. We are encouraged by your recognition of our experiments' importance in elucidating MYT1L's specific role in brain development. In this response, we address each of your comments and concerns point by point, detailing our revisions with line number references. For accurate viewing of these line number references, please use the 'Simple Markup' view in the Track Changes menu. We believe these changes have significantly enhanced our study's rigor and impact, and we hope you will find our revised manuscript suitable for publication.

Reviewer #1 (Remarks to the Author):

This manuscript by Yen and colleagues studies the genomic impact of loss of MYT1L on brain development at single cell resolution in the mouse brain. MYT1L is a transcription factor associated with NDD and prior studies had examined how loss impacts mouse behavior as well as gene expression in bulk RNA-seq. Here, the authors use snRNA-seq at two time points to characterize cell type specific gene expression changes. The authors also integrate their prior CUT&RUN data to determine the direct targets of MYT1L.

Overall, these experiments are important next steps to determine more finely the specific role of this transcription factor in brain development. The use of two time points is important as cell type proliferation and differentiation is dynamic over development.

Specific comments:

1) What is the point of Figure 1G? Not informative and not called out in the text.

We agree with the reviewer that it is not informative and have removed it. We have now expanded the existing adjacent figure (now the new 1J) to include the proportions of nuclei within the different phases of the cell cycle per genotype (lines 127-135) to address Reviewer 2's comment to look at cell cycle proportions.

2) For differential expression, the authors mention "aggregated" clusters. Did the authors recluster specific cell types, clean them up, and then run analysis or simply take the information from the cluster assignments from the original assignment at the all cells level? More details in the methods especially with respect to quality control could be described.

We have clarified and added more detail to the "differential gene expression analysis" section of the methods section (lines 584-604). Additionally, we have added specific quality control metrics and steps to the "snRNAseq data processing" section (lines 560-581).

3) There obviously is no "right" way to perform differential gene expression analysis but the authors use a fairly conservative adjusted p-value of <0.1 . This leads to thousands of genes being called different. I would prefer a slightly more stringent cut-off of 0.05, which is a bit more standard, and see how the lists of genes change. I suspect there will still be hundreds of genes changing that make sense with the biological changes observed.

We appreciate the reviewer's suggestion and have rerun the pseudobulk differential gene expression analysis for E14 and P21 time points with an adjusted p-value threshold of 0.05. This indeed resulted in hundreds of DEGs for E14 and P21 datasets and the overall biological conclusions were unchanged. Additionally, we have added a P1 time point (in response to comments below) and performed the differential expression analysis similarly with a p-value cutoff of 0.05. The figures, text, and methods have been updated accordingly.

4) At E14, the authors mention that the lower layer neurons are the most affected by loss of MYT1L but this might be expected as upper layer neurons are still in the process of being generated and migrating. Moreover, the results at P21 show an increase in the number of lower layer neurons in contrast to the decrease observed at E14. How do the authors reconcile these results?

For E14 and P21 time points, the deep layer neurons had the most differentially expressed genes although their proportions were decreased in E14 and increased in P21. We hypothesize that the slowed maturation of E14 Het deep layer neurons contributes to the decreased proportions relative to WT early on. MYT1L deficiency may lead to a premature fate switch in some progenitors due to altered expression of temporal identity genes that regulate the transition from generating deep to upper layer neurons. This may explain the new observations of a continued decrease in deep layer neurons and increased proportion of upper layer neurons at P1. Intriguingly, at P21, we observe greater proportions of deep layer neurons in the MYT1L Hets, consistent with histology analysis (Chen et al. 2023). We can speculate that this is due to a selective survival advantage during the postnatal wave of apoptosis. Below is a table summarizing the results from previous studies from our lab as well as one from another group. This will now be included as Supplemental Table 6. This will also include D1/D2 neurons, in response to other comments further below.

Study	Observation		Method, n
	D1/D2 neurons	Ratio of Deep to Upper Layers	
Chen et al. Neuron 2021	NA	No differences	Immunofluorescence, n=5
Chen et al. Genome Res 2023	NA	In mutants: more deep layer genes (E14), more deep layer genes (P21), trend towards less upper layer (P21)	Bulk RNAseq GSEA (Figure 1) of Neuron paper data
	NA	In mutants: more Bcl11b (L5/6 marker) density, no change in Pou3f2 (upper layer marker)	Immunohistochemistry, n=6
Weigel et al. Mol Psych 2023	In mutants: more D1/D2	In mutants: more Tbr1, fewer Tbr2, more Reln (L1)	P0 scRNAseq, n=2 per genotype
This study; E14 dataset	In mutants: more D1/D2, but fewer immature D1/D2	In mutants: more Im ExN_3, fewer L5-6 ExN, fewer Im L6 ExN	E14 snRNAseq, n=3 WT, 3 Het, 3 KO
This study; P1 dataset	In mutants: more D1/D2, but not significant	In mutants: more Im L2-4 type 1, fewer Im L2-4 type 2, fewer L5-6 ExN, fewer L6 ExN	P1 snRNAseq, n=8 WT, 4 Het
This study; P21 dataset	In mutants: more D1/D2	In mutants: fewer L2/3 IT ENT, more L2/3 IT PIR-ENTI but not significant, fewer L4/5 IT, more L6 CT, more L6 IT, more L6b CT	P21 snRNAseq, n=6 WT, 6 Het
	In mutants: more Darpp32+ neurons in cortex		Immunofluorescence, n=9 WT, 9 Het

5) I'm not sure I understand the rationale to combine the E14 and P21 data. Those time points are very different in terms of brain development. I think it is better to compare the results of each time point analyzed separately.

We agree that the E14 and P21 time points are indeed very different and difficult to compare directly. Thus, we have conducted an additional experiment to include P1 as an intermediate time point (lines 232-261), bridging the gap between E14 and P21. As suggested, we have analyzed each time point separately for our main analyses. While acknowledging the differences, we believe there is value in examining gene expression patterns across the time points to reveal genotype effects that might be missed when analyzing single time points in isolation. Our pseudobulk differential gene expression analysis across time points identified 63 genes where genotype affects their gene expression pattern across age. These were enriched for negative regulators of TGF-beta signaling, suggesting potential hyper-repression of this pathway, which could impact differentiation and maturation processes (lines 310-320).

6) The authors selected the two timepoints primarily because of the high expression of MYT1L in post-mitotic neurons. The differences between the experimental group and the control group are already striking at the E14 timepoint and continue through P21. The authors also show that the repressor activity of this gene is more prominent and that this effect is dose-dependent. A concern could be that the timepoints are overlooking critical changes at earlier stages of development. This is especially important as the manuscript's introduction states that understanding the timeline could open possibilities for interventions. Such a timeline remains unanswered by these time points, so I suggest the authors describe additional future experiments to further narrow the developmental window.

We appreciate the reviewer's insightful comment and agree that examining additional timepoints could provide a more comprehensive understanding of MYT1L's role throughout neurodevelopment. In response to this valuable feedback, we have expanded our analysis to include the P1 timepoint, which helps bridge the gap between E14 and P21 (lines 232-261). This additional analysis strengthens our current findings and reinforces that MYT1L plays an important role in regulating proportions of neuronal subtypes and loss of MYT1L leads to a disruption of transcriptional maturation. Key findings at P1 include a decreased proportion of deep layer excitatory neurons, a small group of differentially expressed genes that were mostly upregulated in Hets that were associated with developmental processes, and pseudotime analysis revealing that Het cells lag behind WT cells in developmental progression. We observed a progressive increase in DEGs from E14 (5 DEGs) to P1 (89 DEGs) to P21 (413 DEGs) in Hets compared to WT when the time points are analyzed independently (lines 275-276). This temporal progression can suggest two key implications: first, an amplification of indirect effects, where MYT1L's role as a transcriptional regulator could initiate a gradually expanding network of indirect effects on gene expression as development proceeds; and second, there may be a prolonged potential window for postnatal therapeutic interventions, given the continued increase in the effects of MYT1L deficiency after birth (lines 351-355). With this additional data, we then performed an integrative analysis to see how cell classes and cell types are changing across time. We observed elevated expression of TGF-beta signaling regulators, which can have significant consequences on differentiation, maturation, and migration, potentially explaining some of the developmental alterations we observe in MYT1L-deficient animals.

7) The functional connectivity changes are a bit hard to interpret because the within region connectivity was also reduced. Are these results due to more neurons or some other consequence of more neurons affecting connectivity? How do these changes ultimately affect the behavior of the mouse (the authors can speculate)? Are there effects on white matter tracts due to increased numbers of neurons?

Respectfully, it is a bit unclear which section of the paper this comment about functional connectivity might be referring to? We would be happy to try to address it after some clarification from the reviewer/editors.

Reviewer #2 (Remarks to the Author):

The authors are investigating the effect of Myt1l haploinsufficiency or knock-out on different brain cells at different times at single cell resolution. They first show that changes in the doses of Myt1l affects the proportion of different neurons at E14.5, followed by analysis of the transcriptional disruption in excitatory neurons due to loss of Myt1l. They concluded that there is a more immature transcriptional signature upon loss of Myt1l and that the sensitivity of excitatory neurons persists to postnatal development at least till P21. Furthermore, they show that Myt1l acts primarily as a transcriptional repressor during development.

Major revisions:

- At E14.5 layer 6 neurons are the most differentiated and the rest of the postmitotic neurons are just out of intermediate progenitor phase and have not differentiated much, so how can you say layer 6 neurons are the most affected? It would be more interesting to look at P0 as this is the last time point to get KO mice and see the differences with mostly differentiated neurons. Especially as you see more layer 6 neurons at P21 and less upper layers.

We agree that P0 would be a valuable time point for investigating KO animals. However, we must clarify that in our specific MYT1L mouse model, KO animals do not survive beyond birth, making the collection of high-quality tissue for studies challenging, if not impossible—it would require 24-hour continuous monitoring of pregnant dams. This contrasts with the MYT1L KO mouse model described by Wöhr et al. and Weigel et al., where the animals survive for the first few postnatal days, enabling P0 studies on KO animals before their eventual mortality within the first postnatal week.

Nonetheless, recognizing the critical nature of the P0 time point, we conducted an additional experiment analyzing P1 Het animals. Our findings revealed a consistently decreased proportion of deep layer neurons, aligning with our E14 results, as well as altered proportions of upper layer neurons. These observations at P1 provide a crucial link between our E14 and P21 datasets.

- Stain for the neurons that have different proportions to determine if what you see in the snRNA-Seq is also visible in the cortex, especially as the analysis is not in line with your histological analysis in Chen et al. 2021. How do you explain that?

We appreciate the reviewer's suggestion to corroborate our snRNAseq findings with histological analysis. Our refined analysis of progenitor proportions in cell cycle phases at E14 (new Figure 1J) shows good concordance with the immunofluorescence data from Chen et al. 2021. Furthermore, our P21 snRNAseq data aligns well with the histological analysis in Chen et al, 2023, both indicating an increased proportion of deep layer neurons in the MYT1L Het mice.

In addition, an unexpected but consistent finding across E14, P1, and P21 snRNAseq datasets was the increased proportion of D1/D2 striatal neurons in the MYT1L Het cortex. We validated this result through histological analysis on P21 MYT1L Het tissue (Figure 5E-G), confirming our transcriptomic results. The mechanism underlying this increase warrants further investigation but is beyond the scope of this paper.

Finally, while massively parallel barcoding increases snRNAseq throughput, enabling novel insights not possible with in situ methods, greater numbers of nuclei are needed to fully capture the complexity of cortical development. Limitations in cell number and gene dropout inherent to the technology can drive differences between transcriptomic and histological results. We have added text discussing these limitations (lines 406-425), which are applicable to snRNAseq studies in general. Briefly, we think we have learned from this study that future studies measuring cell proportions should increase sample numbers.

- In line with that, Cell cycle disruptions have been shown in Myt1l mice, is there a difference between WT/HET/KO nuclei and their percentage in the cell cycle at E14.5?

We see an increased proportion of RG_2 and InhIP_1 progenitors in the G0/G1 phase in Het and KO cells compared to WT, with a trend towards decreased proportions in the S phase. This is consistent with our previous findings from Chen et al. 2021 that loss of MYT1L may reduce cell proliferation. This additional analysis has been added to Figure 1J and added to the text (lines 129-135).

- Analysis of TF networks using SCENIC would be an interesting add on to get a better sense of the network of Myt1l

Thank you for the suggestion. We have run the SCENIC analysis and have integrated it into the paper (lines 180-186). Briefly, we found that loss of MYT1L does not disrupt the imputed gene regulatory network structure, however key regulons, or modules of TFs and their putative target genes, are differentially expressed mostly in deep layer excitatory neurons. This suggests that MYT1L does not rewire the identified regulons, but rather impacts the expression of 16% of them through TFs that control these regulons.

- There is no analysis on inhibitory neurons but it is implied that the effect is stronger in excitatory neurons, please either perform the analysis and share it or rephrase it

We have added a DEG burden analysis (Figure 2A) for the E14 dataset. Here, we first downsampled and normalized the number of nuclei in each cluster by generating 10 randomly sampled permutations. This is to make each cell type equivalently powered. Then the pseudobulk differential expression analysis was performed for each permutation. The average number of DEGs was reported as the “burden”. This allows us to confirm that the deep layer excitatory neurons had the most DEGs. To further make sure this was not just due to differences in sensitivity due to different rates of gene capture, we confirmed the number of DEGs was not correlated with the number of detected genes ($R^2=0.07$) (lines 146–149).

- Prior bulk RNAseq of the E14 MYT1L Het mouse cortex revealed an immature transcriptional signature when compared to WT. The use of immature in this context seems misleading, in Chen et al. it is discussed that there is earlier neuronal differentiation, which for me suggest faster development than in WT. Please clarify.

We have rephrased our model throughout the text. We now describe it as MYT1L loss causes developing neurons to make the cell fate decision “early”, but then mature slowly after that. We have also changed our language throughout to more often summarize this as ‘disrupted transcriptional maturation’ to reflect this nuance (lines 202-215).

- What are the pathways/GO term enrichments in between WT and HET mice at E14.5? Supplemental tables for DEGs and GO term analysis would be much appreciated.

Interestingly, the number of DEGs when comparing WT and Het mice at E14 is quite low with only 5 DEGs, which is unfortunately too low to conduct any pathway enrichment analyses. We have added supplemental tables for DEGs for all other differential gene expression comparisons. The E14 WT vs KO DEGs are now provided in Supplemental Table 1, the gene dose-responsive DEGs for E14 WT vs KO are provided in Supplemental Table 2, the P1 WT vs Het DEGs are in Supplemental Table 3, and the P21 WT vs Het DEGs are provided in Supplemental Table 4. Hypotheses as to why there are few DEGs when comparing WT and Hets have been added to the text (lines 153-156). We observed a progressive increase in DEGs from E14 (5 DEGs) to P1 (89 DEGs) to P21 (413 DEGs) in Hets compared to WT (lines 275-276). This progression implies an amplification of indirect effects as development and maturation proceeds. We go into more details in lines 348-368.

- Please give examples of the genes that are differentially expressed between WT and HET at P21.

References to differentially expressed genes at P21 have been added to the text (lines 277-281). Additionally, Supplemental Table 5 has been added that lists all DEGs for all cell types.

Minor revisions

- Mytl1 het mice have microcephaly, did you account for that in your analysis at P21?

We appreciate the reviewer's comment. We loaded an equal number of nuclei per genotype for the snRNAseq assay, rather than normalizing by tissue weight. Thus, we are unable to account for overall brain size differences post-hoc, but did make sure we assessed both groups equally. Thus, our analysis primarily reflects the relative proportions of cell types and their transcriptional states within the sampled population, which allows us to compare cellular composition and gene expression between genotypes on a per-nucleus basis.

- I am not entirely sure what this sentence means, please clarify: There were only 11 genes that were not exclusively up or downregulated across all cell types.

This sentence in question was poorly phrased and has been omitted to reduce any confusion.

Reviewer #3 (Remarks to the Author):

In this manuscript, the Dougherty lab continues their examination of MYTL1 deficient mice, this time providing a robust single cell genomics analysis of the developmental effects of Mytl1 abrogation (WT, KO, and Het animals). In general, the results are compelling and the methods development with respect to the calling cards technique is interesting and potentially very valuable although it is only cursorily described. Although their findings are robust and important for better understanding of cortical neuron development, there are also some concerns that need to be addressed in a revised version of the paper.

Major concerns:

1. Related to fig. 1F, it is important to not only show the expression of Myt1l in all cells, but in each genotype separately as well. It is to be expected that the Myt1l knockout (KO) genotype has no expression signal, which serves as a crucial confirmation of the intended mutation.

The engineered mutation in our mouse model causes a frameshift and premature stop codon in MYTL1, resulting in complete protein loss, as confirmed by immunofluorescence staining in E14 KO brain tissue and western blot (Chen et al. 2021). However, the mutant mRNA is still transcribed, leading to similar Myt1l RNA counts in WT, Het, and KO, at least in very sparse and 3' biased data like snRNAseq. Thus, although showing expression of the targeted gene is usually an important QC metric, presenting Myt1l RNA expression in KOs could confuse readers, as our mutation leads to a loss of functional protein but not complete loss of RNA.

In-depth molecular characterization and validation was published in our previous study (Chen et al. 2021) and no additional validations were done in this study. We have included this detail in the methods section (line 438). In this study, we confirmed animal genotypes twice: once with toe or tail tissue at harvest, and another time from the isolated nuclei to confirm there were no sample swaps during processing (lines 452-455).

2. The use of RNA velocity in a single nuclei RNAseq dataset is questionable. Quoting a recent review by the original authors of RNA velocity: "the assumptions such as constant degradation and nuclear export have not been conclusively verified [for single nuclei RNAseq datasets]" (Bergen, V., Soldatov, R. A., Kharchenko,

P. V, & Theis, F. J. (2021). RNA velocity—current challenges and future perspectives. *Molecular Systems Biology*, 17(8), 10282. <https://doi.org/10.15252/MSB.202110282>. If the authors wish to apply this technique, they should demonstrate that their dataset met the prerequisites laid out in this authoritative paper.

We agree with the reviewer about the complications and validity of RNA velocity analysis for this dataset and thus decided to remove the results from this analysis in a previous draft. We apologize for overlooking the last trace of it in Figure 3A and have now removed it.

3. Related to fig. 2C, it is unclear why the authors only performed differential expression (DE) analysis between wildtype (WT) and KO genotypes but did not directly compare WT and heterozygous (Het) genotypes.

This comment is similar to reviewer 2's comment of "What are the pathways/GO term enrichments in between WT and HET mice at E14.5?". Copied from our response above:

Interestingly, the number of DEGs when comparing WT and Het mice at E14 is quite low with only 5 DEGs, which is unfortunately too low to conduct any pathway enrichment analyses. We have added supplemental tables for DEGs for all other differential gene expression comparisons. The E14 WT vs KO DEGs are now provided in Supplemental Table 1, the gene dose-responsive DEGs for E14 WT vs KO are provided in Supplemental Table 2, the P1 WT vs Het DEGs are in Supplemental Table 3, and the P21 WT vs Het DEGs are provided in Supplemental Table 4. Hypotheses as to why there are few DEGs when comparing WT and Hets have been added to the text (lines 153-156). We observed a progressive increase in DEGs from E14 (5 DEGs) to P1 (89 DEGs) to P21 (413 DEGs) in Hets compared to WT (lines 275-276). This progression implies an amplification of indirect effects as development and maturation proceeds. We go into more details in lines 348-368.

4. The methods section is not complete and detailed enough. There is no description of the enrichment analysis procedure. Pseudotime and RNA velocity analyses were not described sufficiently.

Thank you, please note that RNA velocity was removed (see response to Reviewer 3, Comment 2). Description of the enrichment analysis procedure has been added (lines 604-609) and additional details for pseudotime analyses have been added (lines 624-644).

5. Related to fig. 4B, since the authors only performed the single cell experiment workflow once, it is uncertain whether the differences in numbers of each cell type between genotypes accurately reflects real biological changes or simply a result of random sampling.

Because we were concerned about random sampling like this reviewer, we decided to use massively parallel barcoding for our single cell experiments instead of droplet-based methods. While the workflow was only performed once per E14, P1, and P21 time points, the key advantage of massively parallel barcoding is that it enables biological replicates and increased throughput in a cost-effective manner. For example, we had at least 3 biological replicates per genotype per time point. For E14, we had 3 biological replicates per WT, Het, and KO genotypes, resulting in a total of 9 animals. The P1 cohort consisted of 12 animals with 8 WT and 4 MYT1L Het. The P21 cohort consisted of 12 animals, 6 WT and 6 MYT1L Het.

With the current resolution and depth of data, our analysis should be specific, meaning that the differences we observe should be biologically meaningful. However, it may not be optimally sensitive, meaning that there may be other cell types whose differences in proportion could only be gleaned by assaying more nuclei. This limitation in sensitivity is analogous to the challenge single-cell technologies face in detecting differentially expressed genes. We have added text discussing these limitations (lines 403-422).

Minor concerns:

1. Related to fig. 2A: To identify transcriptional changes due to MYT1L deficiency, the authors first aggregated single nuclei data into pseudobulk samples, and then used a method (DESeq2) that is not designed specifically for single cell analysis. The reason for DE analysis after aggregation is unclear, given that DE techniques tailored for single cell datasets without aggregation are readily available.

We appreciate the reviewer's question regarding our choice of pseudobulk analysis for differential expression. Our approach is based on and supported by methodological research, particularly (<https://www.nature.com/articles/s41467-021-25960-2> and <https://www.nature.com/articles/s41467-022-35519-4>). These studies demonstrate that pseudobulk analysis is often optimal for detecting DEGs in single cell studies, especially when dealing with complex experimental designs involving multiple biological replicates. Key advantages are that we get improved statistical power as pseudobulk methods generally outperform single cell specific methods in detecting DEGs, especially for genes with low to moderate expression levels. Pseudobulk methods also account for variability between biological replicates. Methods that do not account for this can identify many differentially expressed genes in the absence of true biological differences across replicates, thus leading to elevated false discoveries.

2. Related to fig. 2A and supplemental fig. 1, when the authors tested overlap between DEGs and SFARI genes, they performed permutation by randomly selecting 932 genes, which is the number of SFARI genes. However, a more conclusive result would arise from sampling 1174 genes from all expressed genes and examine the overlap with SFARI genes each time. Although this may not affect the conclusion, the test needs to be set up reasonably.

We thank the reviewer for catching this. We have redone the analysis by sampling the updated number of DEGs using a more stringent p-value threshold of 0.05 (see Reviewer 1, Comment 3).

3. There is at least one grand statement that may need some qualification. For example on page 4 (I think because there are no page numbers): "This suggests that there are no signals from the differentiating neurons that robustly influence the transcriptional identity of the proliferating progenitor pool at E14." This statement may be most pertinent to L6 neurons (and not all differentiating neurons) and may be affected by the authors' definition of "robustly" and "transcriptional identity."

We apologize for leaving out the page numbers. We have added page and line numbers to the revised document to facilitate references. Further, we have adjusted the language for this particular sentence in question to "Progenitor cells, which do not express MYT1L, showed no DEGs, suggesting that the effects of MYT1L deficiency are intrinsic to MYT1L-expressing cells." (lines 149-150).

Dear editor/reviewers,

Thank you for your thorough review of our manuscript. We appreciate your constructive feedback, which has helped improve the quality and clarity of our work. We are encouraged by your recognition of our experiments' importance in elucidating MYT1L's specific role in brain development. In this response, we address each of your comments and concerns point by point, detailing our revisions with line number references. For accurate viewing of these line number references, please use the 'Simple Markup' view in the Track Changes menu. We believe these changes have significantly enhanced our study's rigor and impact, and we hope you will find our revised manuscript suitable for publication.

Reviewer #1 (Remarks to the Author):

This manuscript by Yen and colleagues studies the genomic impact of loss of MYT1L on brain development at single cell resolution in the mouse brain. MYT1L is a transcription factor associated with NDD and prior studies had examined how loss impacts mouse behavior as well as gene expression in bulk RNA-seq. Here, the authors use snRNA-seq at two time points to characterize cell type specific gene expression changes. The authors also integrate their prior CUT&RUN data to determine the direct targets of MYT1L.

Overall, these experiments are important next steps to determine more finely the specific role of this transcription factor in brain development. The use of two time points is important as cell type proliferation and differentiation is dynamic over development.

Specific comments:

1) What is the point of Figure 1G? Not informative and not called out in the text.

We agree with the reviewer that it is not informative and have removed it. We have now expanded the existing adjacent figure (now the new 1J) to include the proportions of nuclei within the different phases of the cell cycle per genotype (lines 127-135) to address Reviewer 2's comment to look at cell cycle proportions.

2) For differential expression, the authors mention "aggregated" clusters. Did the authors recluster specific cell types, clean them up, and then run analysis or simply take the information from the cluster assignments from the original assignment at the all cells level? More details in the methods especially with respect to quality control could be described.

We have clarified and added more detail to the "differential gene expression analysis" section of the methods section (lines 584-604). Additionally, we have added specific quality control metrics and steps to the "snRNAseq data processing" section (lines 560-581).

3) There obviously is no "right" way to perform differential gene expression analysis but the authors use a fairly conservative adjusted p-value of <0.1 . This leads to thousands of genes being called different. I would prefer a slightly more stringent cut-off of 0.05, which is a bit more standard, and see how the lists of genes change. I suspect there will still be hundreds of genes changing that make sense with the biological changes observed.

We appreciate the reviewer's suggestion and have rerun the pseudobulk differential gene expression analysis for E14 and P21 time points with an adjusted p-value threshold of 0.05. This indeed resulted in hundreds of DEGs for E14 and P21 datasets and the overall biological conclusions were unchanged. Additionally, we have added a P1 time point (in response to comments below) and performed the differential expression analysis similarly with a p-value cutoff of 0.05. The figures, text, and methods have been updated accordingly.

4) At E14, the authors mention that the lower layer neurons are the most affected by loss of MYT1L but this might be expected as upper layer neurons are still in the process of being generated and migrating. Moreover, the results at P21 show an increase in the number of lower layer neurons in contrast to the decrease observed at E14. How do the authors reconcile these results?

For E14 and P21 time points, the deep layer neurons had the most differentially expressed genes although their proportions were decreased in E14 and increased in P21. We hypothesize that the slowed maturation of E14 Het deep layer neurons contributes to the decreased proportions relative to WT early on. MYT1L deficiency may lead to a premature fate switch in some progenitors due to altered expression of temporal identity genes that regulate the transition from generating deep to upper layer neurons. This may explain the new observations of a continued decrease in deep layer neurons and increased proportion of upper layer neurons at P1. Intriguingly, at P21, we observe greater proportions of deep layer neurons in the MYT1L Hets, consistent with histology analysis (Chen et al. 2023). We can speculate that this is due to a selective survival advantage during the postnatal wave of apoptosis. Below is a table summarizing the results from previous studies from our lab as well as one from another group. This will now be included as Supplemental Table 6. This will also include D1/D2 neurons, in response to other comments further below.

Study	Observation		Method, n
	D1/D2 neurons	Ratio of Deep to Upper Layers	
Chen et al. Neuron 2021	NA	No differences	Immunofluorescence, n=5
Chen et al. Genome Res 2023	NA	In mutants: more deep layer genes (E14), more deep layer genes (P21), trend towards less upper layer (P21)	Bulk RNAseq GSEA (Figure 1) of Neuron paper data
	NA	In mutants: more Bcl11b (L5/6 marker) density, no change in Pou3f2 (upper layer marker)	Immunohistochemistry, n=6
Weigel et al. Mol Psych 2023	In mutants: more D1/D2	In mutants: more Tbr1, fewer Tbr2, more Reln (L1)	P0 scRNAseq, n=2 per genotype
This study; E14 dataset	In mutants: more D1/D2, but fewer immature D1/D2	In mutants: more Im ExN_3, fewer L5-6 ExN, fewer Im L6 ExN	E14 snRNAseq, n=3 WT, 3 Het, 3 KO
This study; P1 dataset	In mutants: more D1/D2, but not significant	In mutants: more Im L2-4 type 1, fewer Im L2-4 type 2, fewer L5-6 ExN, fewer L6 ExN	P1 snRNAseq, n=8 WT, 4 Het
This study; P21 dataset	In mutants: more D1/D2	In mutants: fewer L2/3 IT ENT, more L2/3 IT PIR-ENTI but not significant, fewer L4/5 IT, more L6 CT, more L6 IT, more L6b CT	P21 snRNAseq, n=6 WT, 6 Het
	In mutants: more Darpp32+ neurons in cortex		Immunofluorescence, n=9 WT, 9 Het

5) I'm not sure I understand the rationale to combine the E14 and P21 data. Those time points are very different in terms of brain development. I think it is better to compare the results of each time point analyzed separately.

We agree that the E14 and P21 time points are indeed very different and difficult to compare directly. Thus, we have conducted an additional experiment to include P1 as an intermediate time point (lines 232-261), bridging the gap between E14 and P21. As suggested, we have analyzed each time point separately for our main analyses. While acknowledging the differences, we believe there is value in examining gene expression patterns across the time points to reveal genotype effects that might be missed when analyzing single time points in isolation. Our pseudobulk differential gene expression analysis across time points identified 63 genes where genotype affects their gene expression pattern across age. These were enriched for negative regulators of TGF-beta signaling, suggesting potential hyper-repression of this pathway, which could impact differentiation and maturation processes (lines 310-320).

6) The authors selected the two timepoints primarily because of the high expression of MYT1L in post-mitotic neurons. The differences between the experimental group and the control group are already striking at the E14 timepoint and continue through P21. The authors also show that the repressor activity of this gene is more prominent and that this effect is dose-dependent. A concern could be that the timepoints are overlooking critical changes at earlier stages of development. This is especially important as the manuscript's introduction states that understanding the timeline could open possibilities for interventions. Such a timeline remains unanswered by these time points, so I suggest the authors describe additional future experiments to further narrow the developmental window.

We appreciate the reviewer's insightful comment and agree that examining additional timepoints could provide a more comprehensive understanding of MYT1L's role throughout neurodevelopment. In response to this valuable feedback, we have expanded our analysis to include the P1 timepoint, which helps bridge the gap between E14 and P21 (lines 232-261). This additional analysis strengthens our current findings and reinforces that MYT1L plays an important role in regulating proportions of neuronal subtypes and loss of MYT1L leads to a disruption of transcriptional maturation. Key findings at P1 include a decreased proportion of deep layer excitatory neurons, a small group of differentially expressed genes that were mostly upregulated in Hets that were associated with developmental processes, and pseudotime analysis revealing that Het cells lag behind WT cells in developmental progression. We observed a progressive increase in DEGs from E14 (5 DEGs) to P1 (89 DEGs) to P21 (413 DEGs) in Hets compared to WT when the time points are analyzed independently (lines 275-276). This temporal progression can suggest two key implications: first, an amplification of indirect effects, where MYT1L's role as a transcriptional regulator could initiate a gradually expanding network of indirect effects on gene expression as development proceeds; and second, there may be a prolonged potential window for postnatal therapeutic interventions, given the continued increase in the effects of MYT1L deficiency after birth (lines 351-355). With this additional data, we then performed an integrative analysis to see how cell classes and cell types are changing across time. We observed elevated expression of TGF-beta signaling regulators, which can have significant consequences on differentiation, maturation, and migration, potentially explaining some of the developmental alterations we observe in MYT1L-deficient animals.

7) The functional connectivity changes are a bit hard to interpret because the within region connectivity was also reduced. Are these results due to more neurons or some other consequence of more neurons affecting connectivity? How do these changes ultimately affect the behavior of the mouse (the authors can speculate)? Are there effects on white matter tracts due to increased numbers of neurons?

Respectfully, it is a bit unclear which section of the paper this comment about functional connectivity might be referring to? We would be happy to try to address it after some clarification from the reviewer/editors.

Reviewer #2 (Remarks to the Author):

The authors are investigating the effect of Myt1l haploinsufficiency or knock-out on different brain cells at different times at single cell resolution. They first show that changes in the doses of Myt1l affects the proportion of different neurons at E14.5, followed by analysis of the transcriptional disruption in excitatory neurons due to loss of Myt1l. They concluded that there is a more immature transcriptional signature upon loss of Myt1l and that the sensitivity of excitatory neurons persists to postnatal development at least till P21. Furthermore, they show that Myt1l acts primarily as a transcriptional repressor during development.

Major revisions:

- At E14.5 layer 6 neurons are the most differentiated and the rest of the postmitotic neurons are just out of intermediate progenitor phase and have not differentiated much, so how can you say layer 6 neurons are the most affected? It would be more interesting to look at P0 as this is the last time point to get KO mice and see the differences with mostly differentiated neurons. Especially as you see more layer 6 neurons at P21 and less upper layers.

We agree that P0 would be a valuable time point for investigating KO animals. However, we must clarify that in our specific MYT1L mouse model, KO animals do not survive beyond birth, making the collection of high-quality tissue for studies challenging, if not impossible—it would require 24-hour continuous monitoring of pregnant dams. This contrasts with the MYT1L KO mouse model described by Wohr et al. and Weigel et al., where the animals survive for the first few postnatal days, enabling P0 studies on KO animals before their eventual mortality within the first postnatal week.

Nonetheless, recognizing the critical nature of the P0 time point, we conducted an additional experiment analyzing P1 Het animals. Our findings revealed a consistently decreased proportion of deep layer neurons, aligning with our E14 results, as well as altered proportions of upper layer neurons. These observations at P1 provide a crucial link between our E14 and P21 datasets.

- Stain for the neurons that have different proportions to determine if what you see in the snRNA-Seq is also visible in the cortex, especially as the analysis is not in line with your histological analysis in Chen et al. 2021. How do you explain that?

We appreciate the reviewer's suggestion to corroborate our snRNAseq findings with histological analysis. Our refined analysis of progenitor proportions in cell cycle phases at E14 (new Figure 1J) shows good concordance with the immunofluorescence data from Chen et al. 2021. Furthermore, our P21 snRNAseq data aligns well with the histological analysis in Chen et al, 2023, both indicating an increased proportion of deep layer neurons in the MYT1L Het mice.

In addition, an unexpected but consistent finding across E14, P1, and P21 snRNAseq datasets was the increased proportion of D1/D2 striatal neurons in the MYT1L Het cortex. We validated this result through histological analysis on P21 MYT1L Het tissue (Figure 5E-G), confirming our transcriptomic results. The mechanism underlying this increase warrants further investigation but is beyond the scope of this paper.

Finally, while massively parallel barcoding increases snRNAseq throughput, enabling novel insights not possible with in situ methods, greater numbers of nuclei are needed to fully capture the complexity of cortical development. Limitations in cell number and gene dropout inherent to the technology can drive differences between transcriptomic and histological results. We have added text discussing these limitations (lines 406-425), which are applicable to snRNAseq studies in general. Briefly, we think we have learned from this study that future studies measuring cell proportions should increase sample numbers.

- In line with that, Cell cycle disruptions have been shown in Myt1l mice, is there a difference between WT/HET/KO nuclei and their percentage in the cell cycle at E14.5?

We see an increased proportion of RG_2 and InhIP_1 progenitors in the G0/G1 phase in Het and KO cells compared to WT, with a trend towards decreased proportions in the S phase. This is consistent with our previous findings from Chen et al. 2021 that loss of MYT1L may reduce cell proliferation. This additional analysis has been added to Figure 1J and added to the text (lines 129-135).

- Analysis of TF networks using SCENIC would be an interesting add on to get a better sense of the network of Myt1l

Thank you for the suggestion. We have run the SCENIC analysis and have integrated it into the paper (lines 180-186). Briefly, we found that loss of MYT1L does not disrupt the imputed gene regulatory network structure, however key regulons, or modules of TFs and their putative target genes, are differentially expressed mostly in deep layer excitatory neurons. This suggests that MYT1L does not rewire the identified regulons, but rather impacts the expression of 16% of them through TFs that control these regulons.

- There is no analysis on inhibitory neurons but it is implied that the effect is stronger in excitatory neurons, please either perform the analysis and share it or rephrase it

We have added a DEG burden analysis (Figure 2A) for the E14 dataset. Here, we first downsampled and normalized the number of nuclei in each cluster by generating 10 randomly sampled permutations. This is to make each cell type equivalently powered. Then the pseudobulk differential expression analysis was performed for each permutation. The average number of DEGs was reported as the “burden”. This allows us to confirm that the deep layer excitatory neurons had the most DEGs. To further make sure this was not just due to differences in sensitivity due to different rates of gene capture, we confirmed the number of DEGs was not correlated with the number of detected genes ($R^2=0.07$) (lines 146–149).

- Prior bulk RNAseq of the E14 MYT1L Het mouse cortex revealed an immature transcriptional signature when compared to WT. The use of immature in this context seems misleading, in Chen et al. it is discussed that there is earlier neuronal differentiation, which for me suggest faster development than in WT. Please clarify.

We have rephrased our model throughout the text. We now describe it as MYT1L loss causes developing neurons to make the cell fate decision “early”, but then mature slowly after that. We have also changed our language throughout to more often summarize this as ‘disrupted transcriptional maturation’ to reflect this nuance (lines 202-215).

- What are the pathways/GO term enrichments in between WT and HET mice at E14.5? Supplemental tables for DEGs and GO term analysis would be much appreciated.

Interestingly, the number of DEGs when comparing WT and Het mice at E14 is quite low with only 5 DEGs, which is unfortunately too low to conduct any pathway enrichment analyses. We have added supplemental tables for DEGs for all other differential gene expression comparisons. The E14 WT vs KO DEGs are now provided in Supplemental Table 1, the gene dose-responsive DEGs for E14 WT vs KO are provided in Supplemental Table 2, the P1 WT vs Het DEGs are in Supplemental Table 3, and the P21 WT vs Het DEGs are provided in Supplemental Table 4. Hypotheses as to why there are few DEGs when comparing WT and Hets have been added to the text (lines 153-156). We observed a progressive increase in DEGs from E14 (5 DEGs) to P1 (89 DEGs) to P21 (413 DEGs) in Hets compared to WT (lines 275-276). This progression implies an amplification of indirect effects as development and maturation proceeds. We go into more details in lines 348-368.

- Please give examples of the genes that are differentially expressed between WT and HET at P21.

References to differentially expressed genes at P21 have been added to the text (lines 277-281). Additionally, Supplemental Table 5 has been added that lists all DEGs for all cell types.

Minor revisions

- Mytl1 het mice have microcephaly, did you account for that in your analysis at P21?

We appreciate the reviewer's comment. We loaded an equal number of nuclei per genotype for the snRNAseq assay, rather than normalizing by tissue weight. Thus, we are unable to account for overall brain size differences post-hoc, but did make sure we assessed both groups equally. Thus, our analysis primarily reflects the relative proportions of cell types and their transcriptional states within the sampled population, which allows us to compare cellular composition and gene expression between genotypes on a per-nucleus basis.

- I am not entirely sure what this sentence means, please clarify: There were only 11 genes that were not exclusively up or downregulated across all cell types.

This sentence in question was poorly phrased and has been omitted to reduce any confusion.

Reviewer #3 (Remarks to the Author):

In this manuscript, the Dougherty lab continues their examination of MYTL1 deficient mice, this time providing a robust single cell genomics analysis of the developmental effects of Mytl1 abrogation (WT, KO, and Het animals). In general, the results are compelling and the methods development with respect to the calling cards technique is interesting and potentially very valuable although it is only cursorily described. Although their findings are robust and important for better understanding of cortical neuron development, there are also some concerns that need to be addressed in a revised version of the paper.

Major concerns:

1. Related to fig. 1F, it is important to not only show the expression of Myt1l in all cells, but in each genotype separately as well. It is to be expected that the Myt1l knockout (KO) genotype has no expression signal, which serves as a crucial confirmation of the intended mutation.

The engineered mutation in our mouse model causes a frameshift and premature stop codon in MYTL1, resulting in complete protein loss, as confirmed by immunofluorescence staining in E14 KO brain tissue and western blot (Chen et al. 2021). However, the mutant mRNA is still transcribed, leading to similar Myt1l RNA counts in WT, Het, and KO, at least in very sparse and 3' biased data like snRNAseq. Thus, although showing expression of the targeted gene is usually an important QC metric, presenting Myt1l RNA expression in KOs could confuse readers, as our mutation leads to a loss of functional protein but not complete loss of RNA.

In-depth molecular characterization and validation was published in our previous study (Chen et al. 2021) and no additional validations were done in this study. We have included this detail in the methods section (line 438). In this study, we confirmed animal genotypes twice: once with toe or tail tissue at harvest, and another time from the isolated nuclei to confirm there were no sample swaps during processing (lines 452-455).

2. The use of RNA velocity in a single nuclei RNAseq dataset is questionable. Quoting a recent review by the original authors of RNA velocity: "the assumptions such as constant degradation and nuclear export have not been conclusively verified [for single nuclei RNAseq datasets]" (Bergen, V., Soldatov, R. A., Kharchenko,

P. V, & Theis, F. J. (2021). RNA velocity—current challenges and future perspectives. *Molecular Systems Biology*, 17(8), 10282. <https://doi.org/10.15252/MSB.202110282>. If the authors wish to apply this technique, they should demonstrate that their dataset met the prerequisites laid out in this authoritative paper.

We agree with the reviewer about the complications and validity of RNA velocity analysis for this dataset and thus decided to remove the results from this analysis in a previous draft. We apologize for overlooking the last trace of it in Figure 3A and have now removed it.

3. Related to fig. 2C, it is unclear why the authors only performed differential expression (DE) analysis between wildtype (WT) and KO genotypes but did not directly compare WT and heterozygous (Het) genotypes.

This comment is similar to reviewer 2's comment of "What are the pathways/GO term enrichments in between WT and HET mice at E14.5?". Copied from our response above: Interestingly, the number of DEGs when comparing WT and Het mice at E14 is quite low with only 5 DEGs, which is unfortunately too low to conduct any pathway enrichment analyses. We have added supplemental tables for DEGs for all other differential gene expression comparisons. The E14 WT vs KO DEGs are now provided in Supplemental Table 1, the gene dose-responsive DEGs for E14 WT vs KO are provided in Supplemental Table 2, the P1 WT vs Het DEGs are in Supplemental Table 3, and the P21 WT vs Het DEGs are provided in Supplemental Table 4. Hypotheses as to why there are few DEGs when comparing WT and Hets have been added to the text (lines 153-156). We observed a progressive increase in DEGs from E14 (5 DEGs) to P1 (89 DEGs) to P21 (413 DEGs) in Hets compared to WT (lines 275-276). This progression implies an amplification of indirect effects as development and maturation proceeds. We go into more details in lines 348-368.

4. The methods section is not complete and detailed enough. There is no description of the enrichment analysis procedure. Pseudotime and RNA velocity analyses were not described sufficiently.

Thank you, please note that RNA velocity was removed (see response to Reviewer 3, Comment 2). Description of the enrichment analysis procedure has been added (lines 604-609) and additional details for pseudotime analyses have been added (lines 624-644).

5. Related to fig. 4B, since the authors only performed the single cell experiment workflow once, it is uncertain whether the differences in numbers of each cell type between genotypes accurately reflects real biological changes or simply a result of random sampling.

Because we were concerned about random sampling like this reviewer, we decided to use massively parallel barcoding for our single cell experiments instead of droplet-based methods. While the workflow was only performed once per E14, P1, and P21 time points, the key advantage of massively parallel barcoding is that it enables biological replicates and increased throughput in a cost-effective manner. For example, we had at least 3 biological replicates per genotype per time point. For E14, we had 3 biological replicates per WT, Het, and KO genotypes, resulting in a total of 9 animals. The P1 cohort consisted of 12 animals with 8 WT and 4 MYT1L Het. The P21 cohort consisted of 12 animals, 6 WT and 6 MYT1L Het.

With the current resolution and depth of data, our analysis should be specific, meaning that the differences we observe should be biologically meaningful. However, it may not be optimally sensitive, meaning that there may be other cell types whose differences in proportion could only be gleaned by assaying more nuclei. This limitation in sensitivity is analogous to the challenge single-cell technologies face in detecting differentially expressed genes. We have added text discussing these limitations (lines 403-422).

Minor concerns:

1. Related to fig. 2A: To identify transcriptional changes due to MYT1L deficiency, the authors first aggregated single nuclei data into pseudobulk samples, and then used a method (DESeq2) that is not designed specifically for single cell analysis. The reason for DE analysis after aggregation is unclear, given that DE techniques tailored for single cell datasets without aggregation are readily available.

We appreciate the reviewer's question regarding our choice of pseudobulk analysis for differential expression. Our approach is based on and supported by methodological research, particularly (<https://www.nature.com/articles/s41467-021-25960-2> and <https://www.nature.com/articles/s41467-022-35519-4>). These studies demonstrate that pseudobulk analysis is often optimal for detecting DEGs in single cell studies, especially when dealing with complex experimental designs involving multiple biological replicates. Key advantages are that we get improved statistical power as pseudobulk methods generally outperform single cell specific methods in detecting DEGs, especially for genes with low to moderate expression levels. Pseudobulk methods also account for variability between biological replicates. Methods that do not account for this can identify many differentially expressed genes in the absence of true biological differences across replicates, thus leading to elevated false discoveries.

2. Related to fig. 2A and supplemental fig. 1, when the authors tested overlap between DEGs and SFARI genes, they performed permutation by randomly selecting 932 genes, which is the number of SFARI genes. However, a more conclusive result would arise from sampling 1174 genes from all expressed genes and examine the overlap with SFARI genes each time. Although this may not affect the conclusion, the test needs to be set up reasonably.

We thank the reviewer for catching this. We have redone the analysis by sampling the updated number of DEGs using a more stringent p-value threshold of 0.05 (see Reviewer 1, Comment 3).

3. There is at least one grand statement that may need some qualification. For example on page 4 (I think because there are no page numbers): "This suggests that there are no signals from the differentiating neurons that robustly influence the transcriptional identity of the proliferating progenitor pool at E14." This statement may be most pertinent to L6 neurons (and not all differentiating neurons) and may be affected by the authors' definition of "robustly" and "transcriptional identity."

We apologize for leaving out the page numbers. We have added page and line numbers to the revised document to facilitate references. Further, we have adjusted the language for this particular sentence in question to "Progenitor cells, which do not express MYT1L, showed no DEGs, suggesting that the effects of MYT1L deficiency are intrinsic to MYT1L-expressing cells." (lines 149-150).